# Pharmacogenomic screening identifies and repurposes leucovorin and dyclonine as pro-oligodendrogenic compounds in brain repair

Jean-Baptiste Huré[1], Louis Foucault[2], Litsa Maria Ghayad[1], Corentine Marie[1], Nicolas Vachoud [2], Lucas Baudouin [1], Rihab Azmani[2], Natalija Ivljanin[1], Alvaro Arevalo-Nuevo [1], Morgane Pigache[1], Lamia Bouslama-Oueghlani[1], Julie-Anne Chemelle[3], Marie-Aimée Dronne[4], Raphaël Terreux [3], Bassem Hassan [1], François Gueyffier[4], Olivier Raineteau[2] ✉ & Carlos Parras[1] ✉

Oligodendrocytes are critical for CNS myelin formation and are involved in preterm-birth brain injury (PBI) and multiple sclerosis (MS), both of which lack effective treatments. We present a pharmacogenomic approach that identifies compounds with potent pro-oligodendrogenic activity, selected through a scoring strategy (OligoScore) based on their modulation of oligodendrogenic and (re)myelination-related transcriptional programs. Through in vitro neural and oligodendrocyte progenitor cell (OPC) cultures, ex vivo cerebellar explants, and in vivo mouse models of PBI and MS, we identify FDA-approved leucovorin and dyclonine as promising candidates. In a neonatal chronic hypoxia mouse model mimicking PBI, both compounds promote neural progenitor cell proliferation and oligodendroglial fate acquisition, with leucovorin further enhancing differentiation. In an adult MS model of focal de/remyelination, they improve lesion repair by promoting OPC differentiation while preserving the OPC pool. Additionally, they shift microglia from a pro-inflammatory to a pro-regenerative profile and enhance myelin debris clearance. These findings support the repurposing of leucovorin and dyclonine for clinical trials targeting myelin disorders, offering potential therapeutic avenues for PBI and MS.

Preterm birth brain injury (PBI) is the most common cause of death and disability in children under 5 years, affecting 15 million infants yearly born before 37 gestational weeks. Rates of PBI are increasing in developing countries (e.g., 7% in France and the UK, 13% in the US), and are associated with a high level of morbidity and neonatal encephalopathies, leading to persistent cognitive and neuropsychiatric deficits (such as autism spectrum, attention-deficit disorders, and epilepsy)[1–3]. Alterations of oxygen concentrations and perinatal neuroinflammation of PBI dramatically affect the survival and maturation of oligodendrocyte precursor cells (OPCs)[4,5], resulting in hypomyelination, abnormal connectivity, and synaptopathies. Indeed, in both humans and rodents, the perinatal period is a time of active oligodendrogenesis, myelination, and axonal growth within the developing subcortical white matter[6].

[1]Paris Brain Institute, Sorbonne Université, Inserm U1127, CNRS UMR 7225, Hôpital Pitié-Salpêtrière, Paris, France. [2]Univ Lyon, Université Claude Bernard Lyon1, Inserm, Stem Cell and Brain Research Institute U1208, Bron, France. [3]Équipe ECMO, Laboratoire de Biologie Tissulaire et d'Ingénierie (LBTI), UMR5305, Lyon, France. [4]Claude Bernard University, UMR5558 Laboratoire de Biométrie et Biologie Evolutive, CNRS, Villeurbanne, France. ✉e-mail: olivier.raineteau@inserm.fr; carlos.parras@icm-institute.org

Given the severity of long-term consequences and the absence of current treatment, there is an urgent need to develop therapeutic approaches for promoting perinatal brain repair through oligodendrocyte (OL) regeneration.

Another myelin pathology affecting many adults (3 million people) is multiple sclerosis (MS), a chronic autoimmune-mediated disease characterized by focal demyelinated lesions of the central nervous system (CNS)[7,8]. Although current immunomodulatory therapies can reduce the frequency and severity of relapses, they show limited impact on disease progression and are inefficient in lesion repair[9]. Therefore, complementary strategies to protect OLs and induce the generation of new remyelinating OLs are urgently needed to promote repair and eventually end and even reverse MS progression. Accordingly, animal models have demonstrated that remyelination can reestablish fast saltatory conduction velocity[10], prevent axonal loss, and exert neuroprotection[11], resulting in functional clinical recovery[12,13]. Brain remyelination is thought to result mainly from the differentiation of parenchymal OPCs, in regions close to the lateral ventricles, also from adult SVZ-NSC-derived OPCs[14,15], and to a lesser extent from mature OLs[16–19]. The incomplete remyelination in MS patients[20] is attributed to a variety of oligodendroglia-associated intrinsic as well as lesion-environment-related extrinsic factors[21], and the extent of remyelination differs considerably between individual patients and lesion location, declining with disease progression[22]. Despite some factors and small molecules (compounds) being shown to improve myelination in cell cultures, as well as remyelination in animal models, no medication presenting convincing remyelinating capacity in humans has been approved for MS patients so far, with clemastine and analogs of thyroid hormone still having to complete phase III clinical trials[23–26]. Thus, enhancing (re)myelination remains an unmet medical need both in the context of preterm birth brain injury and MS patients.

Owing to their diverse etiologies, onset ages, and potential repair mechanisms, the discovery of new treatments for treating myelin pathologies should aim at globally stimulating endogenous oligodendrogenesis, known to spontaneously increase following injury, at multiple levels, i.e., from the specification of new OPCs from neural progenitor cells (NPCs), to the maturation of pre-existing ones within the parenchyma. A promising drug discovery strategy is to use connectivity maps to identify compounds able to induce transcriptional signatures similar to those observed in a given cell lineage. This approach lies behind the connectivity map initiative, which has been extensively used in various tissues [reviewed in ref. 27]. This strategy has resulted in successful drug repurposing in several contexts [e.g., dietary restriction[28]; ageing and oxidative stress[29,30]], also allowing the discovery of new modes of action for compounds, as well as predicting off-target or side effects.

While most previous studies have followed a gene/pathway candidate approach, we aimed here at using a more comprehensive strategy (Supplementary Fig. 1). To this end, we developed an *in-silico* approach combining both unbiased identification of transcriptional signatures associated with oligodendroglial formation (i.e., transcriptomes) and knowledge-driven curation of transcriptional changes associated with oligodendrogenesis (i.e., OligoScore). We then conducted a pharmacogenomic analysis to identify small bioactive molecules (compounds) capable of globally mimicking this transcriptional signature. Using a scoring strategy based on genes involved in oligodendrogenesis as well as pharmacogenetics/pharmacokinetics criteria, we ranked and selected the most promising compounds. Finally, after validating their pro-oligodendrogenic activity in vitro and ex vivo, we selected the top two compounds, leucovorin and dyclonine, and demonstrated their capacity to promote oligodendrogenesis and myelin repair in vivo, using both neonatal and adult murine models of myelin pathologies.

## Results

### Generation of a transcriptional signature and associated "hub genes" defining oligodendrogenesis

We first set out to generate a broad transcriptional signature of oligodendrocyte lineage cells (Supplementary Data 1 and Supplementary Table 1) by comparing the transcriptome of these cells at different developmental stages, using recently published single-cell- and bulk-transcriptomic datasets from mouse brain cells[31–36]. Given that oligodendroglial cells are robustly generated in mice from the dorsal aspect of the ventricular-subventricular zone (V-SVZ) at neonatal stages[37,38], we integrated the transcriptional signature obtained by comparing dorsal *vs.* lateral neonatal subventricular progenitors (Supplementary Data 1 and Supplementary Table 2)[39,40] to include the transcriptional changes contributing to OL specification from NPCs. Using this strategy, we obtained oligodendroglial transcriptional signatures (Supplementary Data 1 and Supplementary Table 3) enriched either in progenitor cells (1898 genes) or in more mature stages (2099 genes), which we refined to 3372 genes (Supplementary Data 1 and Supplementary Table 4) using further selection criteria ("Methods"). This oligodendroglial signature was indeed enriched in oligodendroglia compared to other neural cells, as illustrated by its expression profile in postnatal brain cells (Fig. 1b) and its enrichment in gene ontology (GO) terms related to OL differentiation, gliogenesis, and glial cell differentiation (Fig. 1c, Supplementary Data 1 and Supplementary Table 4), together with processes linked to neuronal development, synaptic and dendritic organization, known to be modulated by oligodendroglial-dependent brain plasticity and adaptive myelination[41–44].

To identify "hub genes" involved in oligodendrogenesis, we used our oligodendroglial signature to query the SPIED platform (Fig. 1d), which provides a fast and simple quantitative interrogation of publicly available gene expression datasets[45]. This analysis resulted in the identification of hub genes correlated (632 genes) and anticorrelated (623 genes) with our oligodendroglial transcriptional signature (Supplementary Data 1 and Supplementary Tables 5, 6). Correlated hub genes contained numerous key regulators of oligodendrogenesis (including *Olig1, Olig2, Sox8, Sox10, Nkx2-2*), as well as commonly used oligodendroglial markers (such as *Cnp, Mag, Mbp, Mobp, Mog, Opalin, Plp1*; Supplementary Data 1 and Supplementary Table 5). Their expression pattern was enriched in oligodendroglia compared to other brain cells (Fig. 1e), and their associated biological processes were enriched in gliogenesis, CNS myelination, and OL differentiation (Supplementary Data 1 and Supplementary Table 5), thus validating the efficiency of our strategy. Notably, while many correlated hub genes (367/632 genes) were included in our oligodendroglial signature, only a few anti-correlated ones (71/623) were present (Fig. 1e). Interestingly, a large fraction of anti-correlated hub genes (150/632) was expressed in microglia (Fig. 1f) and enriched in GO terms associated with inflammation and immune cell activation (such as IL12, IFNγ or ROS production; Supplementary Data 1 and Supplementary Table 6), known to inhibit OL generation. Altogether, these results indicate that this methodology constitutes an efficient approach to identify a large oligodendroglial transcriptional signature, and to infer hub genes positively or negatively associated with oligodendrogenesis.

### Identification and expert curation of compounds acting on oligodendroglial-associated genes

We then queried SPIED with our oligodendroglial signature to obtain small bioactive molecules (compounds) inducing similar transcriptional changes and found a large number (449) of positively associated compounds. To rank them by the impact of genes involved in oligodendrogenesis, we introduced a knowledge-driven scoring procedure that we named OligoScore (here provided for the community as a

resource; Supplementary Fig. 2; https://oligoscore.icm-institute.org). This curation strategy included more than 430 genes for which loss-of-function and gain-of-function studies have demonstrated their requirement in the main processes of oligodendrogenesis (that we categorized in specification, proliferation, migration, survival, differentiation, myelination, and remyelination). Genes were scored in each process from 1 to 3 (low, medium, strong) either positively or negatively (promoting or inhibiting, respectively), depending on the severity of gain- or loss-of-function phenotypes. For example, *Olig2* and *Sox10*, key regulators of oligodendrogenesis[46–49], have the highest positive values in specification and differentiation, respectively, while *Tnf*[50,51] or *Tlr2*[52] have negative values in processes such as survival and differentiation (Fig. 2a). We validated the efficacy of the OligoScore strategy by querying it with genes differentially expressed in OPCs upon either a genetic (*Chd7* specific deletion in OPCs[53],) or environmental perturbation (systemic injection of the pro-inflammatory cytokine IL1β[54],). Our analysis confirmed that OligoScore accurately predicted the deregulation of oligodendrogenesis-related processes identified in these studies (Supplementary Fig. 3, and "Methods"). Then, using these curated gene sets involved in oligodendrogenesis (Fig. 2b), we interrogated SPIED, finding a large number (393) of positively associated compounds, with 156 in common with those obtained using our broad oligodendroglial signature (Fig. 2c). Focusing on these common 156 compounds, we next used OligoScore to rank them by their total scores (sum of each process' score, defined as pharmacogenomic score) thus selecting those with broader pro-oligodendrogenic activities (Fig. 2d–f, Supplementary Data 1 and Supplementary Table 7). Validating the pertinence of our strategy,

among the top-ranked compounds, we found molecules reported to promote oligodendrogenesis, including liothyronine (thyroid hormone), clemastine[23,55], clomipramine[56], piperidolate[57], luteolin[58], ifenprodil[59], and propafenone[60]. For further analyses, we selected the top 40 compounds ranked by their pharmacogenomic scores and, therefore, by their broader putative pro-oligodendrogenic activities, excluding those already known for their implication in oligodendrogenesis. After studying their pharmacological properties (summarized in Supplementary Data 2 and Methods Table 8), we then selected the top 11 compounds (Sm1-Sm11; Supplementary Data 2 and Methods Table 1) for their pharmacogenomic scores (Fig. 2d), ability to cross the BBB, and low potential toxicity (Fig. 2g, h).

## The pro-oligodendrogenic activity of selected compounds in neonatal neural progenitor cells

To evaluate the activity of selected pro-oligodendrogenic compounds in the context of different neural cell types, we first used mixed neural progenitor cell (NPC) cultures obtained from the neonatal V-SVZ (postnatal day 0 to 1, P0-P1) and amplified using the neurosphere protocol[61,62] (Fig. 3a). We assessed the dose effect of selected compounds to promote NPC differentiation into Sox10+ oligodendroglia using three different concentrations (250, 500, and 750 nM) based on previous reports[23,60,63]. Automatic quantification of Sox10+ cells indicated that all selected compounds presented the strongest effects to promote Sox10+ oligodendroglia at 750 nM (Supplementary Fig. 4). We thus used this concentration for the following in vitro studies. To assess the capacity of compounds to promote NPC differentiation into oligodendroglia, compounds were supplemented in both proliferation

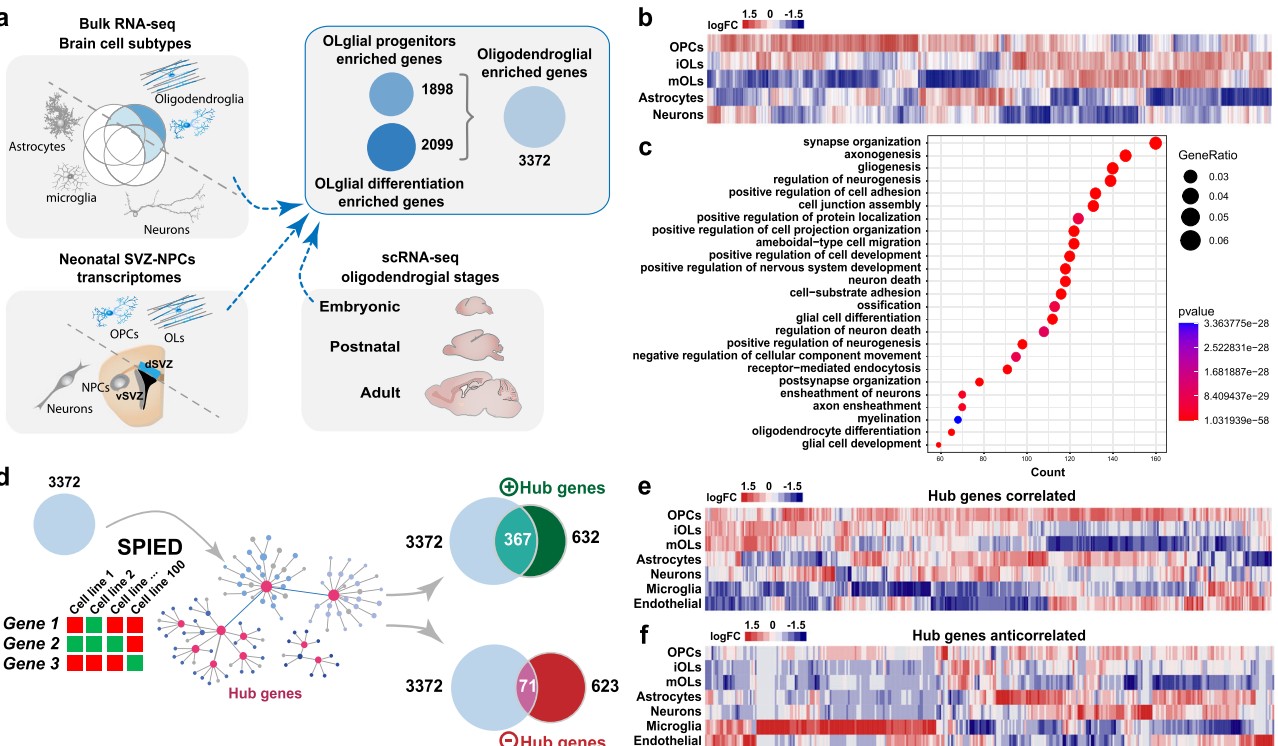

**Fig. 1 | In-silico strategy to generate oligodendroglial-enriched gene sets and their use in pharmacogenomics to identify new hub genes. a** Use of both bulk and single-cell RNA-seq datasets from different brain cell types and oligodendroglial stages, together with the transcriptomic comparison of dorsal (gliogenic) vs. ventral (neurogenic) neonatal forebrain progenitors to identify gene sets enriched either in progenitors, more mature cells, or both, depicted as circles indicating the number of genes. **b** Heat map showing the differential expression of the 3372 genes set in different neural cell types. **c** Top gene ontology biological

processes related to the 3372 genes. **d** Schematics representing the interrogation of the SPIED platform with the 3372 genes to identify hub genes correlated (positive hub genes) or anti-correlated (negative hub genes). Note that most of the positively correlated hub genes are included in the 3372 genes set, whereas only a few of the anti-correlated ones. **e, f** Heatmap visualization of the expression of correlated- (**e**) and anticorrelated-hub genes (**f**) in different brain cells indicates that most of the correlated genes are expressed in oligodendroglia while many anticorrelated hub genes are enriched in microglia, astrocytes, and endothelial cells.

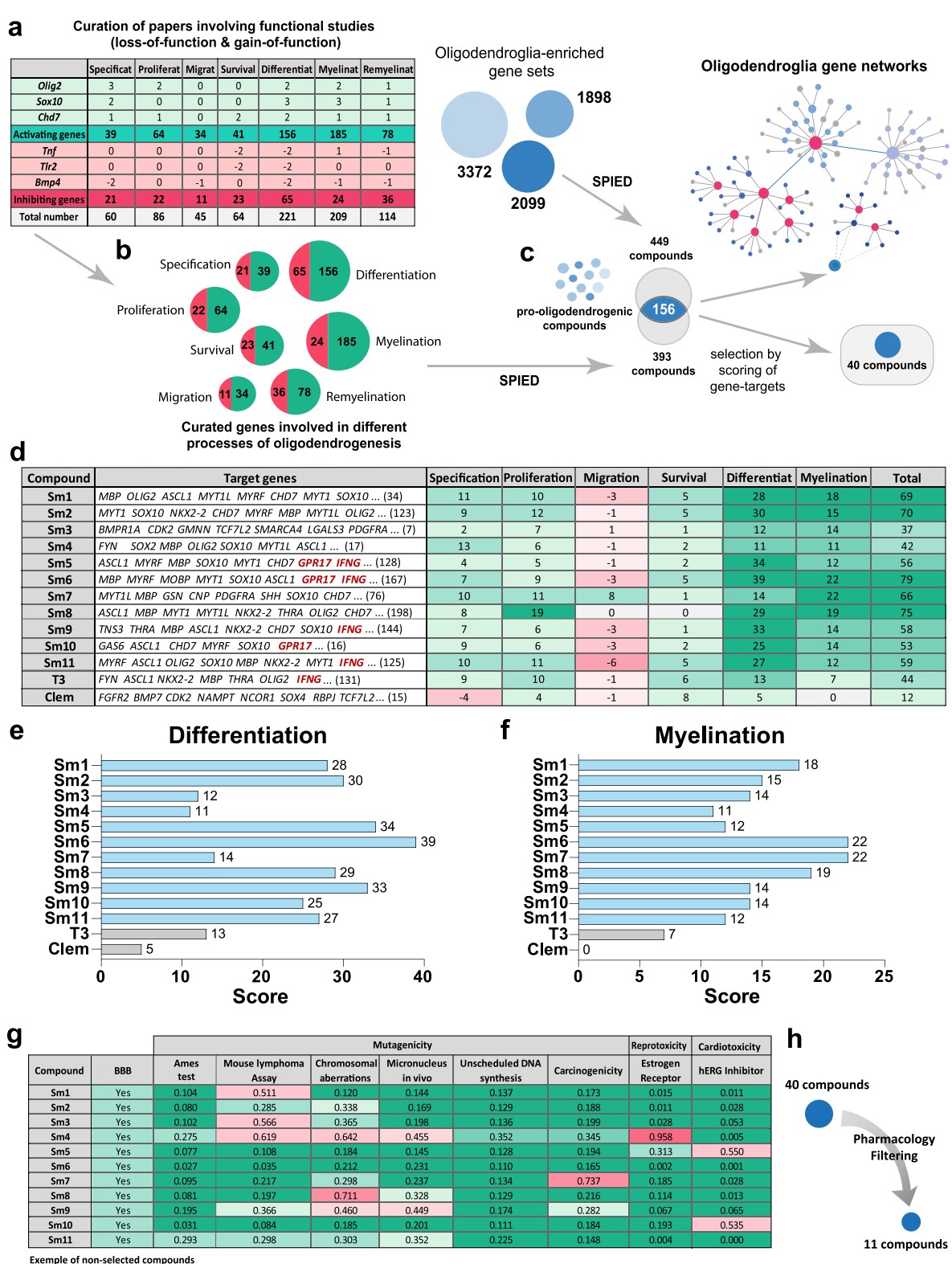

and differentiation media, and clemastine and thyroid hormone were included as positive controls for comparison, given their demonstrated activities on OPC differentiation and myelin formation[23–26]. Remarkably, immunofluorescence labeling of neurons (β-III-tubulin), astrocytes (GFAP), and oligodendroglia (combining PDGFRα and CNP) showed that all 11 selected compounds increased by ~1.5-fold the number of oligodendroglial cells compared to vehicle (DMSO or PBS,

Methods), as illustrated for Sm11 (Fig. 3b, c). In contrast, this increase in oligodendroglia was not found with T3 or clemastine treatment (Fig. 3c), known to only promote OPC differentiation[23,64,65]. Remarkably, the increased number of oligodendroglial cells fostered by our selected compounds was not accompanied by reductions in the number of neurons and astrocytes, nor by changes in overall cell density (Fig. 3d–f). It was rather paralleled by a reduction of

**Fig. 2 | Expert curation ranking and selection of small molecules. a** Table illustrating the scoring of selected genes for their regulation in different oligodendrogenesis processes (positive in green or negative in red), based on bibliographic curation of functional studies. **b** Curated genes subsets promoting (green) or inhibiting (red) each oligodendrogenesis process. **c** Using both curated and oligodendroglial-enriched gene sets to interrogate the SPIED platform and generate a list of small molecules/compounds regulating their expression. Shared compounds were then ranked by the scoring of their curated gene signatures, in order to select 40 compounds with pro-oligodendrogenic transcriptional activity. **d** Table listing the top 11 small molecules (Sm) alongside two positive controls

(T3 and clemastine) exemplifying some of their curated gene targets and their corresponding scoring values (green: positive, red: negative) for each oligodendrogenesis process. **e**, **f** Barplots illustrating the score of selected compounds in the differentiation (**e**) and myelination (**f**) processes, respectively. **g** Table and (**h**) schematics illustrating the pharmacological filtering criteria (BBB permeability, mutagenicity, reprotoxicity, cardiotoxicity, etc.) used to select the top 11 compounds. The green-to-red gradient highlights the probability values from positive to negative activity on each pharmacological parameter. BBB: Blood-Brain-Barrier, hERG: ether-a-go-go related gene potassium channel.

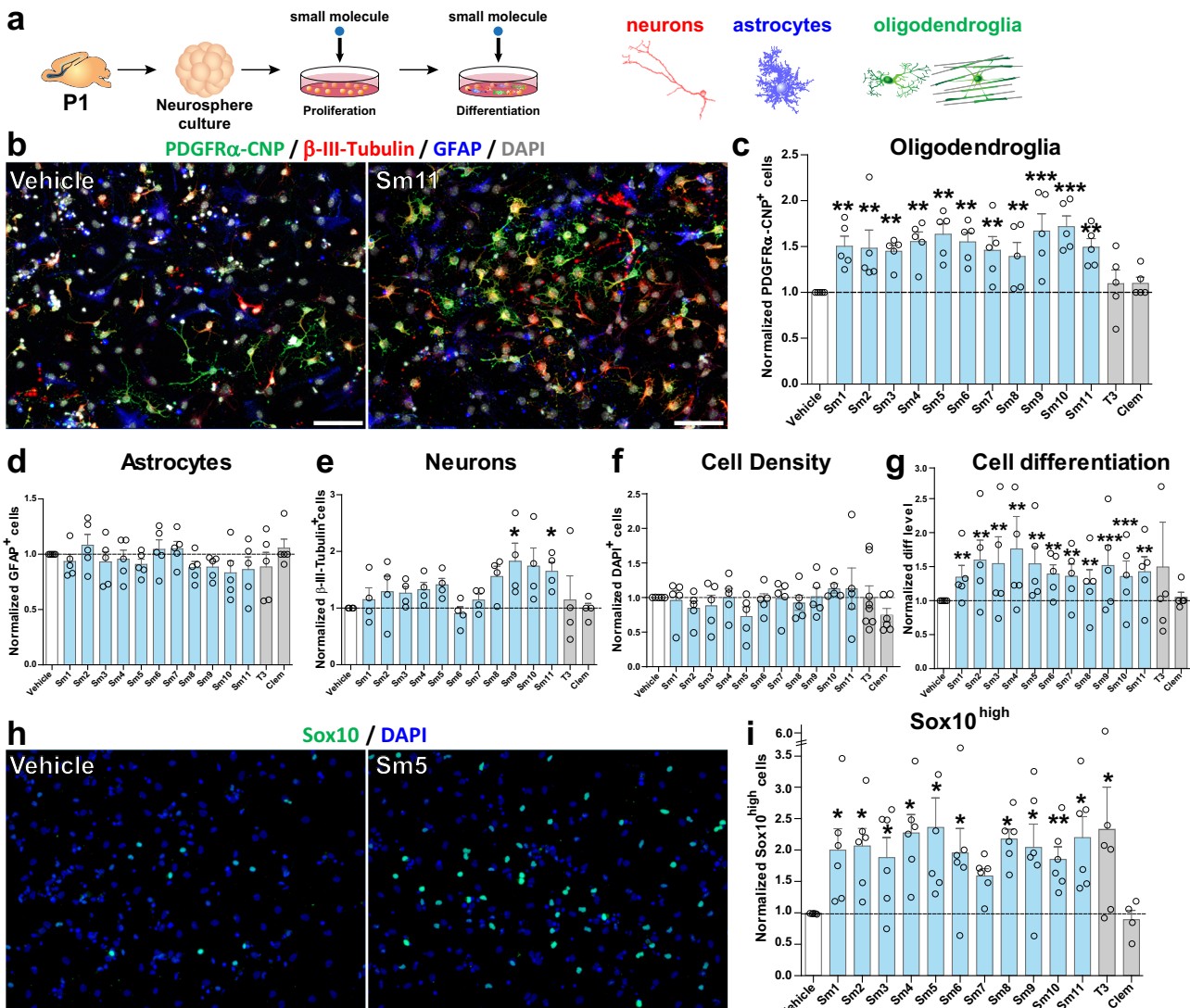

**Fig. 3 | Pro-oligodendrogenic activity of small molecules in neonatal neural progenitor cultures. a** Schematics representing the protocol of neurosphere-derived neural progenitor cell cultures and small molecule administration. **b** Representative images illustrating the immunodetection of neurons (ß-III-tubulin+ cells, red), astrocytes (GFAP+ cells, blue), and oligodendroglia (PDGFRα+ OPCs and CNP+ OLs, green) in vehicle vs. Sm11 treated cultures after 2 days of differentiation. Note the increase of oligodendroglial cells in the Sm11-treated condition. **c**–**g** Quantifications showing the increase of oligodendroglial cells (PDGFRα+ and CNP+ cells) in cultures treated with each of the selected compounds, but not upon T3 and clemastine treatment (**c**), no changes in the number of astrocytes (GFAP+ cells, **d**), neuronal cells (β-III-tubulin+ cells, **e**), or in cell density (**f**), with an increase in cell differentiation (more cells labeled by any of the markers and

less DAPI-only cells) for most selected compounds (**g**). **h** Representative images of the immunofluorescence for Sox10high (iOLs, green) and DAPI (blue) illustrating the increase in treated cultures (Sm5) compared to vehicle cultures after 2 days of differentiation. **i** Quantification showing the increase of Sox10high cells in most compound-treated cultures, including T3. Data was obtained from at least 3 independent experiments, and it is presented as mean ± SEM of fold change normalized to vehicle. Each dot represents a biological replication. Statistical analysis used linear mixed-effects models followed by Type II Wald chi-square tests. *$p < 0.05$; **$p < 0.01$; ***$p < 0.001$. Exact p-values, sample sizes (represented in the dot plots), and source data are provided in the Source Data file and in Supplementary Data 2 - Methods Table 3. Scale bars: (**b**), 20 μm; (**h**), 50 μm.

undifferentiated NPCs (i.e., unlabeled cells; marker negative cells, Fig. 3g), suggesting that selected compounds foster NPC differentiation towards the oligodendroglial fate, with Sm9 and Sm11 also showing a positive effect on neurogenesis (Fig. 3e). To determine their effect on OL differentiation, we quantified the number of immature OLs (iOLs, Sox10[high] cells being PDGFRα[−] and MBP[−]), and found a ~2-fold increase in iOLs (Sox10[high] cells) for T3 and all the selected compounds at 750 nM, as illustrated for Sm5 (Fig. 3h, i and Supplementary Fig. 4). Altogether, these results confirm that our selected compounds have a pro-oligodendrogenic effect in NPC differentiation cultures, without negative impacts on the number of neurons and astrocytes.

## Cell-autonomous effect of selected compounds in OPC differentiation

Given that NPC cultures include several neural cell types, and thus do not allow the evaluation of the direct, cell-autonomous effects of the compounds on OL differentiation/maturation, we turned to primary OPC cultures purified from neonatal (P4) mouse cortices through magnetic cell sorting (MACS). We treated OPCs for three days in the presence of six compounds, for which we did not find any evidence for ongoing research related to oligodendrogenesis (Fig. 4a). Interestingly, automatic quantifications of total cells in each culture (Supplementary Fig. 5) showed that, while no significant changes were seen in the variable density of PDGFRα[+] OPCs (Fig. 4b, c), except for Sm7, both T3, clemastine, and all tested compounds (Sm1, Sm2, Sm5, Sm6, and Sm11) increased the number of MBP[+] OLs compared to vehicle-treated controls (Fig. 4b, d). These results demonstrate the capacity of most selected compounds to cell-autonomously foster OPC differentiation.

## Selected compounds promote oligodendrocyte differentiation and myelination ex vivo in cerebellar explant cultures

Validation of these compounds as a potential treatment for (re)myelination requires further proof of concept before translation into pre-clinical animal models. To this end, we used cerebellar slices[66] to identify the best compounds having pro-oligodendrogenic and pro-myelinating activities ex vivo. This model allowed us to better assess the effect of the selected compounds on OL maturation, myelination, and cytotoxicity in a richer cellular system, also containing immune cells (microglia/macrophages) that are known to influence oligodendrogenesis and (re)myelination[67–73]. The onset of myelination in this model takes place after 7 days in culture (Fig. 5a). Therefore, we incubated the cerebellar explants for three days (7–10 days in vitro, DIV) in the presence of the six compounds showing the highest pro-oligodendrogenic activity in culture (Sm1, Sm2, Sm5, Sm6, Sm7, Sm11), as well as two positive controls (T3 and clemastine), and their associated negative control (vehicle). We analyzed the effect on OL numbers (CC1[+]/Sox10[+] cells) and myelination (CaBP[+] Purkinje cell axons co-labeled with MBP), by immunodetection of all four markers in the same sections. A 'differentiation index' was calculated as the ratio of Sox10[+] cells being CC1[+], and the 'myelination index' as the ratio of CaBP[+] axons being MBP[+] (Supplementary Fig. 6)[66]. Remarkably, all compounds presented an increased differentiation index compared to their negative controls, but only Sm1, Sm2, Sm5, Sm11, and clemastine reached statistical significance (Fig. 5b, c). Moreover, Sm2, Sm5, and Sm11 also induced a robust increase in the myelination index, comparable to the effect of T3 (Fig. 5d, e). Therefore, these results show the pro-oligodendrogenic and pro-myelinating activities of Sm2, Sm5, and Sm11 in cerebellar explant cultures, and together with previous results, demonstrate the pro-oligodendrogenic capacity of these compounds in different brain regions (cerebral cortex and cerebellum). Altogether, based on the global effects of these molecules combining our in vitro and ex vivo experiments (summarized in Supplementary Fig. 6c), and their intended clinical application, we selected Sm5 (dyclonine) and Sm11 (leucovorin) as the best candidates to test their efficiency to promote oligodendrogenesis in pre-clinical models of myelin pathologies.

## Dyclonine and leucovorin promote oligodendroglial regeneration in a mouse model of preterm birth brain injury

Chronic hypoxia is a well-established clinically relevant model of very early preterm birth[74–76]. It is induced by subjecting pups to a low (i.e., 10%) oxygen environment from P3 to P11 (Fig. 6a), which results in diffuse gray and white matter brain injuries, albeit without eliciting a drastic inflammatory response. This period of chronic neonatal hypoxia induces marginal cell death of both neuronal and glial cells along with a delay of OL maturation that persists into adulthood[77]. We assessed the effects of dyclonine (Sm5) and leucovorin (Sm11), as pro-oligodendroglial treatments, by performing intranasal administration of these compounds immediately following the period of hypoxia (i.e., from P11 to P13, see Methods; Fig. 6a and Supplementary Fig. 7a). This strategy represents a very promising noninvasive technique for drug administration, that was already demonstrated to impact brain cell behavior and eventually their proliferation[40,75]. To examine the impact of dyclonine and leucovorin on proliferation, brains were analyzed at P13, corresponding to the end of the treatment period, after EdU administration one hour before sacrifice to label cells in S-phase. Quantification of EdU[+] cell density within the dorsal V-SVZ revealed an overall increase in proliferation following hypoxia, with no marked additive effects of the treatment (Fig. 6b, c). Notably, the proportion of Olig2[+] cells labeled with EdU was significantly increased upon dyclonine and leucovorin treatment (Fig. 6d, e), suggesting an increase in the proliferation of oligodendroglial committed progenitors. Furthermore, the proportion of EdU[+] cells expressing Olig2 was increased by dyclonine and leucovorin treatment (Fig. 6f), suggesting that these compounds also promote the oligodendroglial fate acquisition from NPCs, in line with their pro-oligodendrogenic activity in neonatal NPC cultures (Fig.3). We then investigated the capacity of dyclonine and leucovorin to rescue the reduction in OL differentiation and maturation induced by hypoxia[75]. To this end, OL density and maturation were analyzed within the cortex six days following the end of the treatment, at P19, when the rate of cortical OL maturation and myelination is highest. While the density of oligodendroglial cells (Olig2[+] cells) was similar between normoxic (Nx; control condition) and hypoxic groups, with no major effects attributable to compounds administration (Supplementary Fig. 7b, c), their differentiation into CC1[+]/Olig2[+] OLs was impaired by hypoxia but this difference with normoxic group was rescued by both dyclonine and leucovorin treatments, with leucovorin also reaching statistical difference with the hypoxic group (Fig. 6g, h). We next assessed the effect of hypoxia and treatments on the number of myelinating OLs (mOLs), identified as Olig2[+] cells expressing GSTπ, a marker restricted to mOLs [[78]; Fig. 6i, j, and Supplementary Fig. 7d, e]. Quantification of Olig2[+]/GSTπ[+] cell density revealed a marked effect of hypoxia on OL maturation that was fully rescued by leucovorin treatment. In order to substantiate these observations, we extended this analysis by automatically quantifying Olig2[+]/GSTπ[+] cells to other forebrain regions. At this age, mOLs showed the highest density in ventral brain regions such as the thalamus and hypothalamus when compared to dorsal regions such as the isocortex (Fig. 6k). Further, normalized Olig2[+]/GSTπ[+] cell densities on Nx mice (Fig. 6l) revealed a stronger effect of hypoxia on OL maturation in ventral brain regions. We confirmed these qualitative observations by performing a manual quantification within the hypothalamus (Fig. 6I). Those confirmed a stronger effect of hypoxia on OL maturation in the hypothalamic lateral zone, compared to the cortical region, as well as a full rescue by leucovorin treatment (Fig. 6i–l). In contrast, in this region again, dyclonine treatment did not increase Olig2[+]/GSTπ[+] cell density when compared to hypoxic

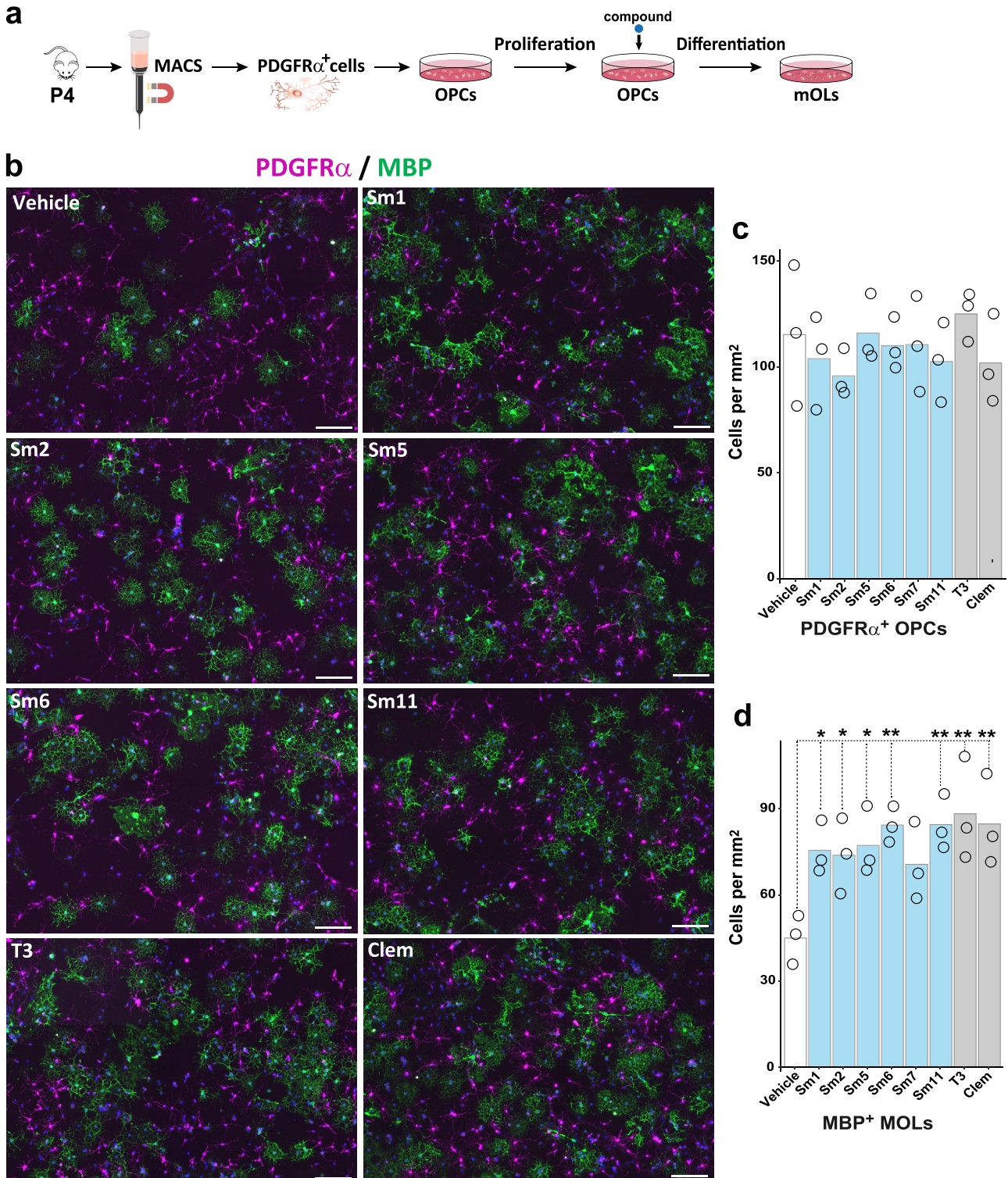

**Fig. 4 | Cell-autonomous effect of selected compounds in OPC differentiation.**
**a** Schematic representing the protocol of OPC purification, culture, and compound treatment. **b** Representative images illustrating the immunodetection of OPCs (PDGFRα⁺ cells, magenta) and differentiating OLs (MBP⁺ cells, green).
**c**, **d** Quantification of the number of OPCs (PDGFRα⁺ cells, **c**) and number of OLs (MBP⁺ cells, **d**) per mm² in different treated conditions, showing that most compounds present a significant increase in the number of differentiating OLs

compared to the vehicle treatment. Data are presented as mean +/− SEM from 3 independent experiments. Each dot represents a biological replication. $N = 3$. Statistics were performed using One-way ANOVA to compare the cell counts across different treatments, followed by Dunnett's test to compare each treatment with Vehicle (control group). *$p < 0.05$; **$p < 0.01$; ***$p < 0.001$. Exact $p$-values and source data are provided in the Source Data file. Scale bars: 20 μm.

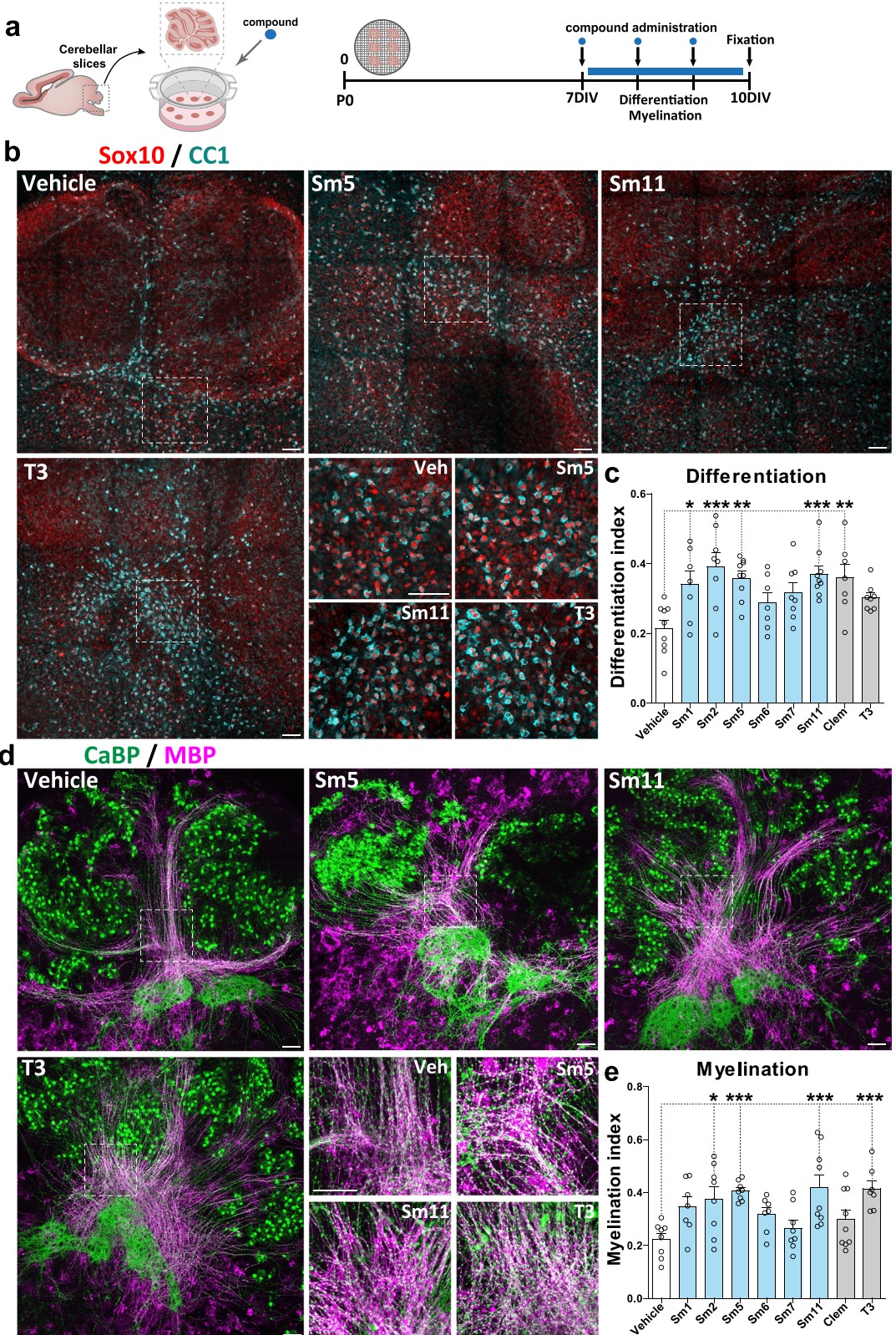

mice (Fig. 6k, l). We finally investigated the effect of treatments on MBP immunoreactivity within this ventral region (Fig. 6k). Whereas MBP levels have been shown to be reduced following hypoxia at P11 in several brain regions (Foucault et al., 2024; Chourrout et al., 2022), our results show a return to baseline by P19 in the hypothalamus (Supplementary Fig. 7f, g). Analyses of MBP optical density and Olig2$^+$/GSTπ$^+$ cell densities suggest an increase of MBP expression

per mOL following dyclonine treatment, while the MBP/mOLs ratio was back to normal in leucovorin-treated animals (Fig. 6m). These results, in line with our previous in vitro and ex vivo results, confirm the therapeutic capacity of dyclonine and leucovorin to promote oligodendrogenesis in a model of preterm birth brain injury and validate the intranasal approach to administer these compounds and their capacity to cross the blood-brain barrier (BBB). Further, our

**Fig. 5 | Selected compounds promote oligodendrocyte differentiation and myelination ex vivo in cerebellar explant cultures. a** Schematic illustrating the protocol of the cerebellar explant culture model and timing of compound administration. **b** Images of explants illustrating the effects of compounds (Sm5, Sm11, and T3) on oligodendrocyte differentiation by immunodetection of Sox10[+] oligodendroglia (red) and CC1[+] OLs (blue). **c** Quantification of the differentiation index (SOX10[+]CC1[+]/SOX10[+] cells) shows an increase following treatment with Sm1, Sm2, Sm5, Sm11, and clemastine compounds. **d** Immunofluorescence of explants illustrating compound effects (Sm5, Sm11, and T3) on myelination of Purkinje axons by immunodetection of CaBP[+] axons/cells (green) and MBP[+] (pink). **e** Quantification of the myelination index (surface MBP[+] CaBP[+]/ surface CaBP[+]) showing an increase following treatment with Sm2, Sm5, Sm11, and T3 compounds. Data are presented as mean ± SEM. Each dot represents a biological replication. Statistical unpaired bilateral Wilcoxon Mann Whitney test. *$p < 0.05$; **$p < 0.01$; ***$p < 0.001$. Exact $p$-values, sample sizes (represented in the dot plots), and source data are provided in the Source Data file and in Supplementary Data 2 - Methods Table 3. Scale bars: 100 μm.

results show a superior capacity of leucovorin to promote OL maturation restoring normal myelinating OL numbers following injury.

### Dyclonine and leucovorin promote OPC differentiation while maintaining the OPC pool in a mouse model of adult demyelination

To further assess the pro-oligodendrogenic capacity of dyclonine (Sm5) and leucovorin (Sm11) in vivo in the context of myelin pathologies occurring later in life, we used the mouse model of adult focal de/remyelination induced by lysolecithin (LPC) injection into the corpus callosum (Fig. 7a). Compounds were administered via drinking water (see "Methods" section), in line with oral administration protocols previously used in adult mice and humans[79–82]. Notably, no alterations in the drinking behavior of treated mice were observed, ensuring the expected intake of compounds (Supplementary Fig. 8b). We first analyzed the lesions at 7 days post-lesion (7 dpl) induction, when newly formed OLs started to remyelinate the lesion[83,84]. The lesion area was identified by the high cellular density (DAPI staining), the abundance of microglia/macrophages (Iba1[+] cells), and the reduction of myelin content (using myelin oligodendrocyte glycoprotein, MOG) (Supplementary Fig. 8c). We determined the effect of the compounds to promote oligodendrogenesis at the lesion site by combinatory immunodetection of Olig2, CC1, and Olig1, allowing to distinguish OPCs (Olig2[high]/CC1[-]/Olig1[nuclear-cyto] cells) and three stages in OL differentiation[53,84]: immature OL (iOL) 1 (iOL1, Olig2[high]/CC1[high]/Olig1[-] cells), iOL2 (Olig2[high]/CC1[high]/Olig1[high-cyto] cells), and mOL (Olig2[low]/CC1[low]/Olig1[low] cells; Fig. 7b, c). First, we found that dyclonine, and to a greater extent, leucovorin, increased the number of oligodendroglial cells (Olig2[+] cells) in and around the lesion (Fig. 7c–e₁, f). Quantification of oligodendroglial stages showed that while the administration of these compounds did not change the density of OPCs (Olig2[high]/CC1[-]/Olig1[nuclear-cyto] cells) within the lesion area (Fig. 7g), both compounds increased the number of immature OLs. Specifically, leucovorin increased the number of iOL1s (Fig. 7c, h), while both leucovorin and dyclonine increased iOL2s (Fig. 7c, i). Given that both compounds increased the number of differentiating OLs without diminishing the pool of OPCs in the lesion area, we looked at the proliferative status of OPCs, using the Mcm2 proliferation marker (Fig. 7i–l), and found that both compounds promote a two-fold increase in the density of proliferative OPCs (Fig. 7m). Finally, calculation of the differentiation ratio (number of iOL2 per OPC) showed that whereas this ratio was 1 OL for 4 OPCs in the vehicle-treated lesions, it increased by 2 folds in dyclonine-treated (2 OLs for 4 OPCs) and by more than 3-fold in leucovorin-treated (3 OLs for 4 OPCs) lesions (Fig. 7n). We then performed an additional experiment to assess for possible improved effects using higher doses of the compounds (10-fold for leucovorin and 2-fold for dyclonine) and to compare their efficacy with that of clemastine, a pro-myelinating compound[23]. At either dose, we observed a similar increase in oligodendroglial (Olig2[+] cells) and immature OL (iOL1 and iOL2) densities, suggesting that we had reached an optimal dose for eliciting the pro-oligodendrogenic effects of our compounds in this model (Supplementary Fig. 8d–h). Moreover, while clemastine induced an increase in iOL2 density

similar to dyclonine and leucovorin (Supplementary Fig. 8h), both leucovorin and dyclonine additionally increased the density of proliferating OPCs (Fig. 7m, o), an effect that was absent for clemastine (Fig. 7o).

### Dyclonine and leucovorin accelerate oligodendrocyte formation and remyelination in a mouse model of adult demyelination

To obtain further proof that the increase in newly formed OLs contributes to accelerated remyelination, we fluorescently labeled adult OPCs and their progeny (OPCs and OLs) by administering tamoxifen to *Pdgfra-CreER^T; Rosa26^{stop-YFP}* mice for five consecutive days before inducing the LPC lesion (Fig. 8a). At 10 dpl, we identified by immunofluorescence newly formed OLs (YFP[+] cells) being either Bcas1, a marker of immature/pre-myelinating OLs[85], or GSTπ, a marker restricted to myelinating OLs[78]. Interestingly, we did not find overlap between YFP[+]/Bcas1[+] cells and YFP[+]/GSTπ[+] cells, suggesting that, indeed, Bcas1 and GSTπ identify immature/pre-myelinating and mature/myelinating OLs, respectively. This analysis showed that, similar to clemastine, dyclonine and leucovorin increased more than 2-fold the number of newly formed myelinating OLs (YFP[+]/GSTπ[+] cells) in the lesion area (Fig. 8b, d) without significant reduction in the lesion volume at this time point likely due to lesion variability (Fig. 8c). Performing electron microscopy imaging, we confirmed by ultrastructural analysis the increase in myelinating OLs (identified by their round- or oval-shape nucleus having densely packed chromatin and processes wrapping around myelinated axons; Supplementary Fig. 9b) in animals treated with leucovorin, dyclonine, and clemastine compared to vehicle-treated animals (Fig. 8e). Moreover, quantification of myelinated axons in the lesion area showed a tendency to decrease the g-ratio of axons in leucovorin-, dyclonine-, and clemastine-treated animals compared to vehicle-treated controls, suggesting an increased remyelination (more wrapping) induced by the compound treatment, that reaches significance in the case of leucovorin treatment for axons larger than 1 μm (Fig. 8f, g). Altogether, these results show that similar to clemastine, leucovorin, and dyclonine promote OL differentiation, thus remyelination, in vivo in the context of adult brain demyelination, with leucovorin showing the strongest effect. The possibility of leucovorin and dyclonine directly inducing myelin formation would require further investigation. Moreover, the increased number of oligodendroglial cells in the lesion area, together with the increase in OPC proliferation, indicate that in the context of adult demyelinating lesions, these compounds are capable of promoting OL differentiation while maintaining the pool of OPCs by fostering their proliferation, an effect not found with clemastine (Fig. 7o).

### Dyclonine and leucovorin accelerate myelin clearance and microglial transition from pro-inflammatory to pro-regenerative profiles in a mouse model of adult demyelination

Finally, given the recognized pro-remyelinating properties of microglia[86] and the capacity of our compounds to repress inflammatory genes such as Interferon-gamma (Fig. 2d), we investigated their potential anti-inflammatory and pro-regenerative activity mediated by microglia/macrophages (Iba1[+] cells). We assessed various microglial profiles, including phagocytic (CD68[+]/Iba1[+] cells), pro-inflammatory (Cox2[+] and iNOS[+] cells), and pro-regenerative (Arg1[+]/Iba1[+] cells)

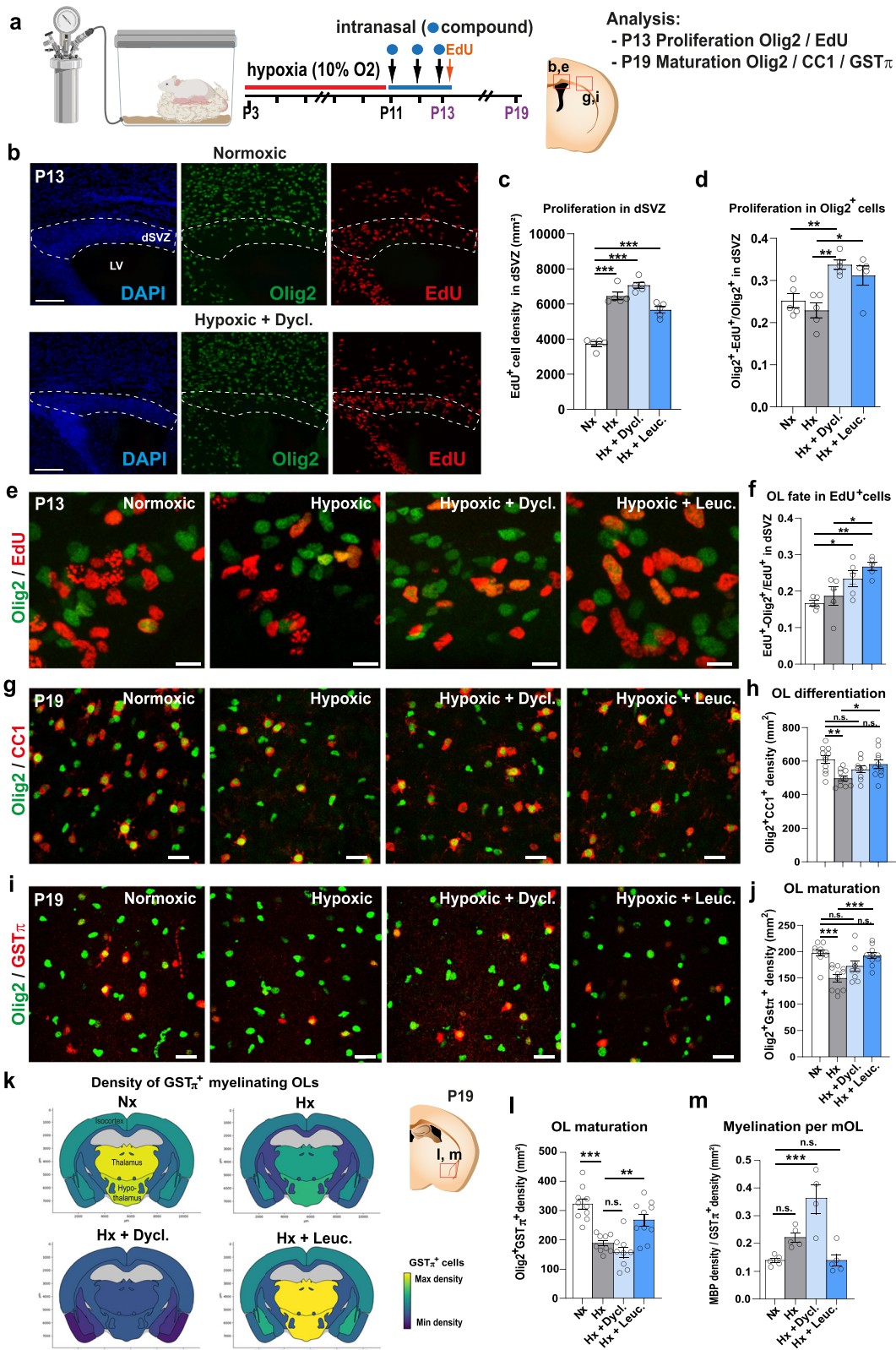

microglia, known to follow the dynamics of lesion repair in this model[67,68]. Interestingly, within the lesion area at 7 dpl, there was a reduction in the fraction of phagocytic microglia/macrophages (CD68+ cells; Fig. 9a, c) and a reduction in the area of myelin debris labeled by dMBP[73] accompanied by an increased proportion of the dMBP phagocyted by CD68+ cells (Fig. 9a, d, e), suggesting faster myelin clearance. Furthermore, we found a decrease in Cox2+ and iNOS+ pro-

inflammatory microglial profiles (Fig. 9f–h), paralleled with a strong increase in pro-regenerative microglial phenotype (Arg1+ cells; Fig. 9i, j), suggesting that both dyclonine and leucovorin promote an earlier transition between these microglia/macrophage profiles.

In summary, these results demonstrate that in the context of adult brain de/remyelination, oral administration of dyclonine and leucovorin promotes the generation of newly formed OLs while preserving

**Fig. 6 | Dyclonine and leucovorin promote oligodendroglial regeneration in a mouse model of preterm birth brain injury. a** Schematic illustrating the workflow used to assess the capacity of dyclonine and leucovorin to promote OPC proliferation and rescue OL maturation following neonatal chronic hypoxia. **b** Images of dorsal SVZ at P13, delimited by DAPI counterstaining, showing Olig2 and EdU immunodetection in brain sections of control animals (i.e., normoxic) and following hypoxia and dyclonine treatment. **c** Quantification of EdU$^+$ cell density (i.e., proliferative cells) in the dorsal SVZ at P13, illustrating the increase in proliferation observed following hypoxia with or without treatment. **d**–**f** Olig2/EdU immunodetection and corresponding quantifications showing that dyclonine and leucovorin increase (**d**) the ratio of proliferative OPCs (Olig2$^+$EdU$^+$ cells) and (**f**) the OL fate of EdU$^+$ cells in the dorsal SVZ at P13. **g** Olig2 and CC1 immunodetection and quantification (**h**) showing that dyclonine and leucovorin rescue the reduced density of differentiating OLs (CC1$^+$ cells) induced by neonatal chronic hypoxia within the cortex at P19, compared to the density found in the normoxic group, with leucovorin reaching statistical difference with the hypoxic group. **i** Olig2 and GSTπ immunodetection and quantification (**j**) showing that only leucovorin rescues the density of myelinating OLs (GSTπ$^+$ cells) following hypoxia within the cortex at P19. **k** Heatmap representations depicting quantifications of Olig2$^+$/GSTπ$^+$

cell density in distinct regions of P19 coronal brain sections. Note the reduced density induced by hypoxia (Hx) in most brain regions with a pronounced effect in the thalamus and hypothalamus, compared to normoxic (Nx) control brains, and the rescue of Olig2$^+$/GSTπ$^+$ cell density following leucovorin (Hx + Leuc.) but not dyclonine (Hx + Dycl.) treatment. **l** Quantification of Olig2$^+$/GSTπ$^+$ cells in the hypothalamic lateral zone of P19 animals confirming that leucovorin, but not dyclonine, treatment rescues the reduced density of hypoxic animals. **m** Quantification of the ratio between MBP immunodetection (shown in Fig. 6 extended data **f**, **g**) and GSTπ$^+$ cell density showing an increased myelination per mOL in dyclonine-treated animals. Dyc., dyclonine; Leuc., leucovorin; Data are presented as Mean ± SEM. Statistics were performed using One-way ANOVA to compare the cell counts across different treatments, followed by Dunnett's test to compare each treatment with either Normoxia (Nx) or Hypoxia considered as control groups. Each dot represents a biological replication. *$p < 0.05$; **$p < 0.01$; ***$p < 0.001$, n.s., non-significant. Exact $p$-values, sample sizes (represented in the dot plots), and source data are provided in the Source Data file and in Supplementary Data 2 - Methods Table 3. scale bars: 100 μm in **b**, 10 μm in **e**; 20 μm in (**g**) and (**i**).

the OPC pool size. Furthermore, in addition to their direct effects on OLs, these compounds also appear to improve lesion repair by accelerating myelin clearance and promoting the transition from pro-inflammatory to pro-regenerative microglial profiles.

## Discussion

To date, no medication demonstrating convincing remyelination efficacy in humans has been approved for treating myelin pathologies, including preterm-birth brain injuries (PBI) and multiple sclerosis (MS). While most previous studies have followed a gene/pathway candidate approach, here we used a more comprehensive strategy to fill this gap (Fig. 10). Leveraging transcriptomic datasets through a pharmacogenomics analysis and developing an expert curation of genes previously involved in oligodendroglial biology (provided as a resource for the scientific community: OligoScore, https://oligoscore.icm-institute. org), we identified and ranked small bioactive molecules (compounds) for their capacity to foster transcriptional programs associated with various aspects of oligodendrogenesis, including OPC proliferation, differentiation, and (re)myelination. We then demonstrated the pro-oligodendrogenic activity of some of these compounds in mice using several approaches: i.e., (1) in vitro, in cultures of neural progenitor cells and primary oligodendroglial progenitors, (2) ex vivo, in cerebellar explant cultures, and (3) in vivo, using both a model of perinatal chronic hypoxia and a model of adult focal demyelination with spontaneous remyelination. Compounds repurposed to promote remyelination in MS currently in phase III clinical trials, such as clemastine and thyroid hormone analogs[23,55,87], generate OLs at the expense of OPCs, thus potentially depleting OPCs in the long term. Remarkably, the compounds we identified here, leucovorin and dyclonine, promote the generation of new OLs while maintaining the pool of OPCs stable by inducing their concomitant proliferation and differentiation, an effect not found using clemastine (Fig. 7). The dual capacity of leucovorin and dyclonine to promote at the same time OPC proliferation and differentiation in the context of adult remyelination, is supported by our pharmacogenomics analysis predicting that both leucovorin and dyclonine induce the expression of *Olig2, Ascl1, Sox2, and Myt1* (Supplementary Data 2 and Supplementary Tables 7, 11, 12), key transcription factors promoting both proliferation and differentiation programs (Nielsen et al., 2004; Ligon et al., 2007; Nakatani et al., 2013; Zhang et al., 2018; Vue et al., 2020). Finally, in the context of an adult demyelination model, leucovorin and dyclonine also improve lesion repair by promoting in parallel a microglia-mediated myelin debris clearance, and a faster transition from pro-inflammatory to pro-regenerative microglial profiles. Therefore, these compounds represent promising candidates for a more

comprehensive and sustained regenerative response in various oligodendroglial pathologies, with leucovorin showing the largest beneficial effects in all conditions tested here.

Selected for their broad transcriptional effects, leucovorin and dyclonine may increase oligodendrogenesis and (re)myelination by targeting multiple cell types and mechanisms. Concerning targeted cell types, our results obtained in neonatal neural progenitor cultures and in the neonatal hypoxia model in vivo indicate that leucovorin and dyclonine act onto neural progenitor cells to promote their differentiation into oligodendroglial cells. Second, using purified OPC primary cultures, we demonstrate that they can act cell-autonomously in OPCs to promote their differentiation/maturation, in agreement with a recent study showing that folic acid, also involved in folate metabolism, can enhance oligodendrocyte differentiation[88]. Third, in the in vivo context of a neonatal hypoxia model of PBI, both compounds can promote oligodendroglial proliferation, with only leucovorin efficiently restoring the number of myelinating OLs (Fig. 6). It is interesting to note that whereas dyclonine did not recover normal myelinating OL density, myelination however returned to baseline as revealed by optical density measurement of MBP expression. This suggests that dyclonine did not impede compensatory myelin production by the remaining OLs. The long-term consequences of this incomplete recovery and the possible increase in myelination capacity of individual OLs upon dyclonine treatment would need further investigation. Fourth, in the in vivo model of adult focal demyelination with spontaneous remyelination, both leucovorin and dyclonine accelerate the generation of newly formed and remyelinating OLs, while maintaining the pool of OPCs by increasing their proliferation (Figs. 7 and 8). This capacity to maintain the pool of adult OPCs, not found using clemastine, has important implications for the long-term treatment of MS patients, whose remyelination capacity decreases with age and disease progression[89,90]. Finally, leucovorin and dyclonine also target microglia/macrophages, accelerating their phagocytosis of myelin debris and their transition from pro-inflammatory (Cox2$^+$ and iNOS$^+$ cells) to pro-regenerative (Arg1$^+$/Iba1$^+$ cells) profiles. These effects, which have also been described in clemastine treatments[91,92], are known to favor OPC differentiation and remyelination in this model[67–73,93].

Dyclonine and leucovorin have been approved by the FDA in 1955 and 1952, respectively. Dyclonine has been used as an oral anesthetic administered in throat lozenges and is known to inhibit Nav1.8, a voltage-dependent Na channel primarily expressed on small, unmyelinated peripheral sensory neurons, most of which are nociceptors[94]. Recent findings identified dyclonine's ability to enhance synaptic activity in cultured hippocampal mouse neurons, with chronic

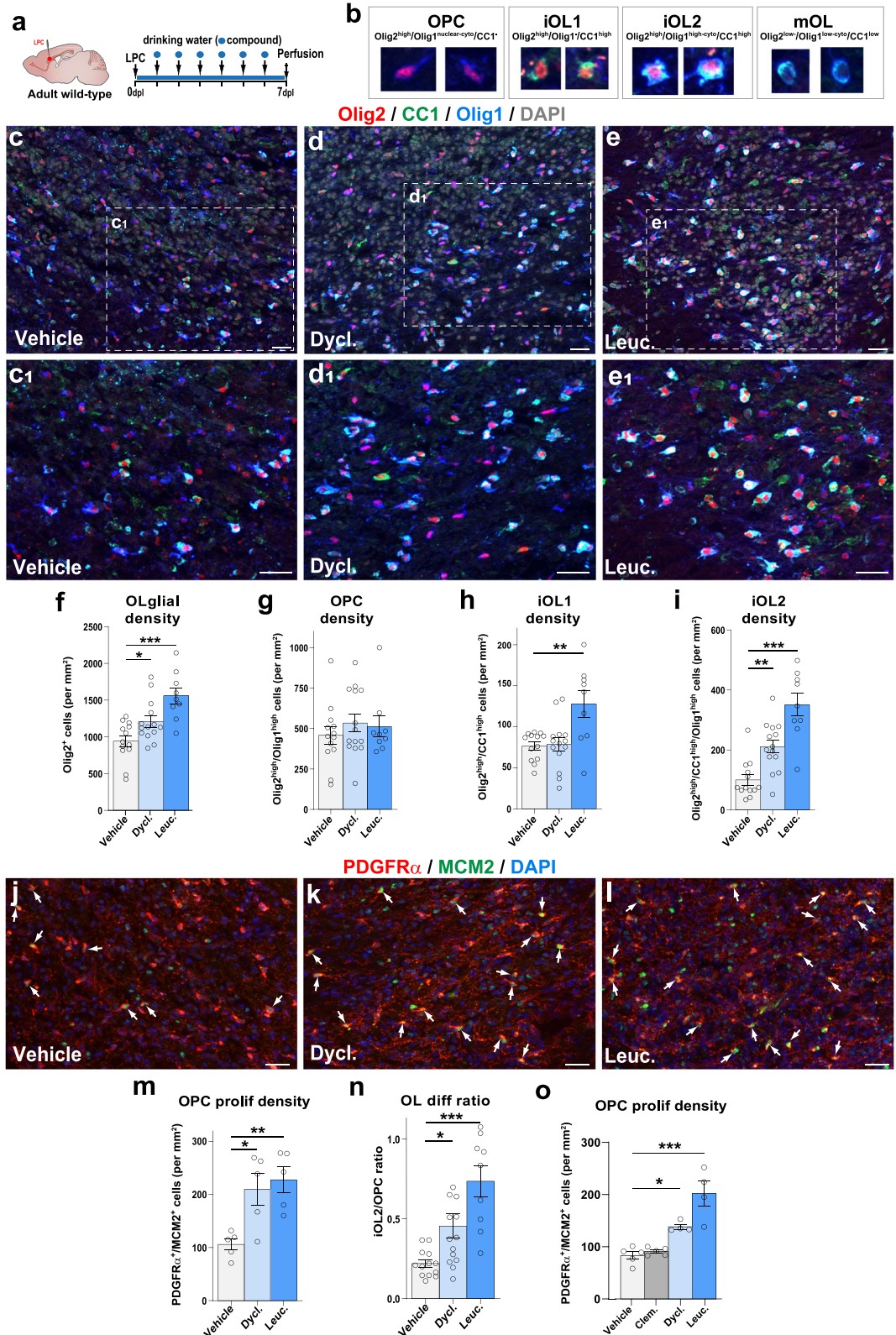

treatment of mice over several months in their drinking water leading to increased respiration and ATP production in brain mitochondria[95]. In a study of drug repositioning in the context of Friedreich Ataxia[79], dyclonine was found to confer protection against diamide-induced oxidative stress through binding to the transcription factor NRF2 (nuclear respiratory factor 2) and activation of the NRF2 pathway[96]. Finally, dyclonine may also mediate pro-oligodendrogenic effects

through its suggested regulation of different hormones, including estrogen receptors, thyroid hormone receptor β, and the androgen receptor[95] which represent possible therapeutic targets in MS[97].

Leucovorin (aka folinic acid or folinate), is a naturally occurring folate (from Latin *folia*, foliage), a general term designing molecularly-related metabolites participating in the folate cycle, a metabolic pathway that, depending on cell requirements, can lead to nucleotide

**Fig. 7 | Dyclonine and leucovorin promote both OPC proliferation and differentiation in a mouse model of adult demyelination. a** Schematics illustrating the protocol for LPC demyelination in the corpus callosum, the timing of compound administration in drinking water and analysis. **b** Images illustrating the 4 oligo-dendroglial stages obtained combining Olig2/CC1/Olig1 immunofluorescence: OPCs (Olig2$^{high}$/Olig1$^{high}$), iOL1 (Olig2$^{high}$/CC1$^{high}$), iOL2 (Olig2$^{high}$/CC1$^{high}$/Olig1$^{high}$) and mOLs (Olig2$^{low}$/CC1$^{low}$/Olig1$^{low}$). **c–e$_1$** Representative images of Olig2/CC1/Olig1 immunofluorescence in the lesion site (depicted by the high cellularity with DAPI) in control mice (**c,c$_1$**, $N = 13$), dyclonine-treated (**d,d$_1$**, $N = 14$), and leucovorin-treated (**e,e$_1$**, $N = 9$) representative of the quantifications shown in (**f–i**). (**f–i**) Quantification showing the increase in the density of Olig2$^+$ oligodendroglia in dyclonine- and leucovorin-treated lesions (**f**), no changes in OPC density (**g**), the increase in iOL1 density only in leucovorin treated lesions (**h**), and the increased iOL2 density in dyclonine- and leucovorin-treated lesions (**i**). (**j–l**) Representative images showing the increase in proliferation (Mcm2$^+$ cells, green) of OPCs (PDGFRα$^+$ cells, red) in the lesion area in dyclonine-treated (**k**) and leucovorin-

treated (**l**) compared to vehicle controls (**j**) ($N = 5$ per group). The white arrow indicates proliferating OPCs (PDGFRα$^+$/Mcm2$^+$ cells). **m** Quantification of the pro-liferating OPC density showing a 2-fold increase in the compound-treated groups. **n** Quantification of the OL differentiation ratio showing 2 to 3 times differentiation increase in dyclonine- and leucovorin-treated lesions, respectively. **o** Quantification of the proliferating OPC density in a replicated experiment comparing with clem-astine, showing that contrary to clemastine, both dyclonine and leucovorin increase the OPC proliferation density in the lesion. Dycl., dyclonine; Leuc., leu-covorin; Clem., clemastine. Data are presented as Mean ± SEM. Each dot corre-sponds to a biological replicate. Statistics were performed using One-way ANOVA to compare the cell counts across different treatments, followed by Dunnett's test to compare each treatment with Vehicle (control group). *$p < 0.05$; **$p < 0.01$; ***$p < 0.001$. Exact p-values, sample sizes (represented in the dot plots), and source data are provided in the Source Data file and in Supplementary Data 2 - Methods Table 3. Scale bars: 20 µm.

synthesis, mitochondrial tRNA modification, or methylation, respec-tively impacting proliferation, mitochondrial respiration, and epige-netic regulation[98]. Abnormal folate metabolism has been causally linked to a myriad of diseases, but it is often unclear which biochemical processes and cellular functions are affected in each disease (Zheng and Cantley, 2018). Molecularly different from folic acid, a synthetic folate, leucovorin has a different entrance and metabolism in the folate cycle. Both leucovorin and folic acid can be transformed by different enzymes into 5-methyl-tetrahydro-folate, which efficiently crosses the BBB using the folate receptor alpha (FRα) transporter, and is thought to be the main active folate in the CNS[99,100]. Leucovorin has the advantage over folic acid of using other transporters present in the choroid plexus to get into the CNS, i.e., the proton-coupled folate transporter (PCFT) and the reduced folate carrier (RFC)[101]. Moreover, folic acid must be metabolized by dihydrofolate reductase (DHFR) to enter the folate cycle, with this reaction being slow and easily reaching saturation, thus limiting the therapeutic use of high doses of folic acid[99,100]. For these reasons, leucovorin has been used to treat epileptic patients having mutations in the folate receptor alpha gene, FOLR1[101], as well as in the context of cancer chemotherapy to decrease the toxic effects of methotrexate, an inhibitor of the DHFR. Our results showing that leucovorin increases oligodendrogenesis and (re)myelination both in vitro and in vivo suggest that it promotes these processes at least in part by fostering high levels of folate metabolism. Moreover, the current clinical administration of leucovorin to adults for large periods guarantees its long-term administration to MS patients with-out major side effects. Finally, some studies point out that molecules involved in the folate cycle can reduce the inflammatory response by either inhibiting TNFα, IL-1β, or iNOS-dependent NO production[102,103]. These anti-inflammatory effects parallel our pharmacogenomics strategy, indicating that leucovorin downregulates the gene coding for Interferon gamma (*IFNG*), a key inflammatory cytokine, and our results in adult remyelinating lesions indicating that leucovorin acts in microglia/macrophages accelerating the transition from pro-inflammatory to pro-regenerative profiles. Altogether, our current understanding of dyclonine and leucovorin activities warrants the broad pro-regenerative activities found in our mouse models of myelin pathologies, with superior effects of leucovorin. Nevertheless, future studies will be required to elucidate the specific mechanisms of action of these compounds in oligodendroglial development, remyelination, and lesion repair.

In conclusion, our study underscores pharmacogenomic analysis as an efficient and low-cost approach to identifying pro-oligodendrogenic compounds. By developing a scoring system for genes implicated in oligodendrogenesis (OligoScore), we could rank the most promising compounds to demonstrate their pro-oligodendrogenic and pro-myelinating activities using both in vitro and ex vivo murine culture systems. This resulted in the selection of

leucovorin and dyclonine, two FDA-approved compounds, which we demonstrated to have brain repair and pro-oligodendrogenic activities in both preterm birth brain injury and adult demyelination mouse models, with leucovorin demonstrating broader beneficial effects. This work paves the way to clinical trials repurposing these compounds to promote brain repair in the context of both early- and late-life-onset myelin pathologies such as PBI and MS.

## Limitations of the study

Despite the cellular (neural precursor cells, oligodendroglia, and microglia) and molecular mechanisms (folate cycle and anti-oxidative stress metabolism) underlying the beneficial effects of leucovorin and dyclonine in oligodendrogenesis and myelin repair shown here, follow up studies are crucial to unravel the particular mechanisms underlying the effects of each compound in each pathological context, and in cell types, including neurons and astrocytes. For clinical translation in treating preterm birth brain injuries, the capacities of leucovorin and dyclonine to rescue other aspects of preterm birth brain injury, such as inflammation and malnutrition, will need further investigation. Also, the increase in OPC proliferation and numbers mediated by dyclonine in the hypoxia model do not rescue the density of myelinating OLs, contrasting with its effects in adult brain de/remyelination, calling for a better understanding of the environmental differences of each pathological model, such as microglial involvement. The proposed increased myelination capacity per OL (e.g., increased number of internodes) mediated by dyclonine treatment in the hypoxia model will require direct experimental evidence. Finally, further studies are needed to explore the capacity of leucovorin and dyclonine to pro-mote myelin repair in the context of the aging brain, which is only obtained with clemastine or T3 treatment when combined with metformin (Neumann et al., 2019a). Thus, considering the possible synergy between leucovorin and dyclonine with metformin in the context of the aging brain constitutes an essential follow-up pre-clin-ical research before starting clinical trials in the context of MS lesion repair.

## Methods
### Data processing and Oligodendroglial gene enrichment
Data Selection: transcriptomic datasets were selected from single-cell and bulk-transcriptomic studies generated in neural cell types during mouse development at the embryonic, postnatal, and adult stages[31–36]. Supplementary Data 2 - Methods Table 2 summarizes the datasets used, criteria for selecting DEGs, p-values, fold-changes, GSE numbers, and data details. For most of the datasets, enriched genes were iden-tified using supplementary tables provided in the publications, selecting DEGs (≥1.5-fold change) or genes with enriched expression ($p < 0.05$). Detailed methods for specific dataset analyses are provided below.

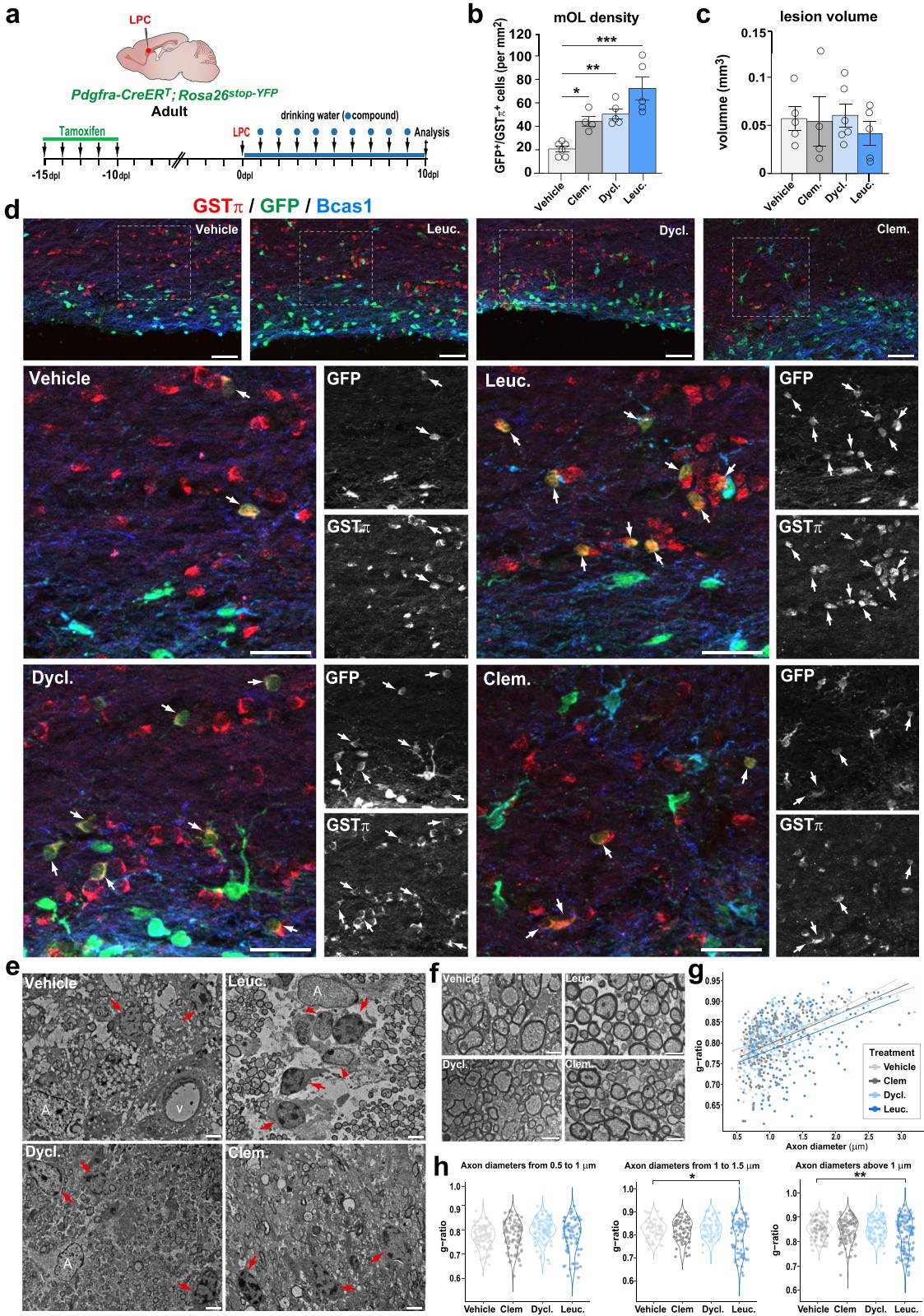

**Postnatal dorsal NSCs and TAPs.** we used data from Raineteau's lab (GSE60905) focusing on neural stem cells (NSCs) and transit amplifying precursors (TAPs) isolated at P4 from the dorsal and lateral SVZ regions, using transgenic mice and specific markers for each cell type (i.e., *Hes5::GFP* and *prominin-1* mice for NSCs and *Ascl1::GFP* mice for TAPs). The microarray data were processed in two phases (see 'dNSCs/ TAPs' script): Normalization: Using R and the 'affy' package, data were

normalized with the Robust Multiarray Averaging (RMA) method[104]. Differential Analysis: Limma analysis[105] identified DEGs with a filter of FC > 1.2 and $p < 0.05$.

**Oligodendrocyte lineage in postnatal brain.** we used data from Ben Barres' lab (GSE52564), and using RNA-Seq from purified cells, we identified oligodendrocyte-specific genes. Cells were isolated at P7

**Fig. 8 | Leucovorin and dyclonine accelerate oligodendrocyte formation and remyelination in a mouse model of adult demyelination. a** Schematics illustrating the protocol used for tracing OPCs and newly formed OLs by YFP-reporter induction prior to LPC demyelination by tamoxifen mediated Cre-recombination of a stop cassette in *Pdgfra-CreER^T; Rosa26^stop-YFP* mice, followed by compounds' administration in the drinking water and analysis at 10 days post-lesion (dpl). **b** Quantification of the density of newly formed (GFP⁺) myelinating OLs (GSTπ⁺), showing that dyclonine and leucovorin, like clemastine, increase the generation of remyelinating OLs compared to vehicle. **c** Quantification of the lesion volume indicating no major reduction at 10 dpl in treated animals. **d** Representative images of adult-generated YFP⁺ immature OLs (Bcas1⁺ cells, blue) or myelinating OLs (GSTπ⁺ cells, red). **e** Electron microscopy images at 10 dpl illustrating the increase in mOLs (red arrows) identified by their typical ultrastructural traits (round- or oval-shape nucleus having densely packed chromatin and processes wrapping around axons presenting compact myelin traits) in compounds' treated lesions compared to controls (Vehicle). Note that leucovorin-treated lesions have more frequent iOLs (arrowheads) in the lesion area, characterized by showing less densely packed

chromatin. **f** Representative micrographs illustrating remyelinated axons in the lesion area in different treatments. **g** Scatter plot representing the quantification of myelin sheath thickness (g-ratio) per axon diameter in the lesion area of vehicle- (light gray), clemastine- (gray), dyclonine- (light blue), and leucovorin-treated (blue) animals indicating a tendency to increase myelin thickness (lower g-ratios) in compound-treated animals compared to vehicle, with leucovorin-treated animals showing the strongest effect. **h** Violin plots quantifications of g-ratios in axons with different thicknesses indicating a significant decrease in g-ratio (thicker myelin) of axons above one micrometer in the lesion area of leucovorin-treated animals. A, astrocytes. V, vessel. Clem., clemastine; Dycl., dyclonine; Leuc., leucovorin. Data are presented as Mean ± SEM. Each dot corresponds to a biological replicate. Statistics were performed using One-way ANOVA to compare the cell counts across different treatments, followed by Dunnett's test to compare each treatment with Vehicle (control group). *$p < 0.05$; **$p < 0.01$; ***$p < 0.001$. Exact $p$-values, sample sizes (represented in the dot plots), and source data are provided in the Source Data file and in Supplementary Data 2 - Methods Table 3. Scale bars: (**d**) 20 μm; (**e, f**) 1 μm.

---

(astrocytes and neurons) and P17 (oligodendrocyte lineage cells). Filtering criteria included FPKM > 0.5 to remove low-expression genes and a final filter for overexpressed/repressed genes in the oligodendrocyte lineage (FC > 1.2–9.2). The '*dNSCs/TAPs*' transcriptional signature was combined with the '*OL lineage*' transcriptional signatures to obtain lists of intersection genes ('*OL specification*' signature) and exclusion genes ('*OL differentiation*' signature) (see results section).

**Single-cell RNA-seq Oligodendroglial Clusters**. using Marques and colleagues (2016) supplemental table 'aaf6463 Table S1', sheet 'specific genes', we pooled all genes except those of the VLMC column, obtaining 532 unique genes. To extract more information, raw data from 5072 cells were processed with the Seurat package v2 (Stuart et al., 2019) using the following criteria: min.cells = 10; min.genes = 500; mitochondrial percentage < 0.10, 20 dimensions to find neighbors, a resolution of 0.9 to find clusters, selecting the top 150 genes from each cluster from FindAllMarkers function, identifying 1598 DEGs (Supplementary Data 2 - Methods Table 2).

**Pharmacogenomics using the SPIED platform**
The transcriptional signatures obtained were used to query the CMAP 2.0 database via the wSPIED platform. The CMAP 2.0 (CMAP2.0) database consists of transcriptional profiles corresponding to the effects of small molecules at various concentrations and treatment times on panels of human cell lines (~100). This enables the identification of therapeutic molecules capable of inducing transcriptional changes similar to those observed during oligodendrogenesis. Query files were simple text files with columns containing gene IDs and their enrichment/repression (+1 or −1, respectively). The output table contained identifiers of small molecules (correlative and anti-correlative), ranked by Z-score (Pearson-type regression analysis), and $p$-value.

**Gene ontology and cell-specific enrichment analysis**
GO analysis of selected genes was performed using the Cluster Profiler R-package (2012), Metascape website[106], EnrichR website[107], Panther website[108].

**Expert curation and scoring of genes regulating oligodendrogenesis**
We implemented a knowledge-driven scoring procedure by curating over 1000 publications involving functional studies demonstrating the role of specific genes in oligodendrogenesis (OPC specification, proliferation, migration, survival, differentiation, myelination, and remyelination) under physiological or pathological conditions. This curation implicated 430 genes, scored for each process from 1 to 3 (low, medium, strong), either positively or negatively, based on their

function in the process. To share our expert curation scoring strategy, we established a user-friendly platform OligoScore (OligoScore platform), evaluating the regulatory impact on transcriptional programs involved in oligodendrogenesis and (re)myelination.

**Validation of the OligoScore strategy**
We validated the OligoScore strategy using genes enriched in specific oligodendroglial stages and genes deregulated in OPCs upon perturbations. First, we queried OligoScore with the top 2000 genes enriched in OPCs and myelinating oligodendrocytes (mOLs), obtained by comparing their transcriptomes from purified brain cells (RNAseq[36]) and sorting them by their expression ratio between OPCs and mOLs, respectively. Using oligodendroglial single-cell transcriptomes[31,32], we performed a similar analysis using genes enriched in OPCs (723 genes, logFC > 0.5 in OPC and cycling OPC clusters versus mOL1/2 clusters) and in mOLs (1428 genes, logFC > 0.5 in mOL1/2 clusters versus OPC and cycling OPC clusters). Results confirm expectations, showing that genes enriched in OPCs are involved in several processes of oligodendrogenesis (specification, proliferation, differentiation, and (re)myelination; Supplementary Fig. 3a, c), while genes enriched in mOLs are mainly involved in differentiation and myelination (Supplementary Fig. 3b, d). Second, we queried OligoScore with genes differentially expressed in OPCs upon genetic (P7 Chd7-deleted OPCs, $p$-value < 0.01; Marie et al., 2018), or environmental perturbation (P10 OPCs in the systemic IL1β-mediated neonatal neuroinflammatory model, FDR < 0.05; Schang et al., 2022). This identified the affected processes in agreement with these studies: reduced survival and differentiation of Chd7-iKO OPCs (Supplementary Fig. 3e), and increased proliferation and reduced differentiation of P10 OPCs in the IL1β-model (Supplementary Fig. 3f). In addition, this analysis provided mechanistic insights, identifying genes responsible for the deregulation of each oligodendroglial process. Gene sets are used to query OligoScore and the full results are provided in the Excel file named 'Supplementary Data 3 and OligoScore validation tables'.

**Hub genes and small molecules list generation**
To obtain either hub genes or small molecules, we interrogated the SPIED3 version (a searchable platform-independent expression database III; SPIED3; GeneHubs and CmapG function). To generate a list of hub genes, we uploaded gene symbols corresponding to the large oligodendroglial gene subset (3372 genes), whereas for small molecules, we used gene symbols corresponding to either the large oligodendroglial gene sets (1898, 2099 and 3372 genes) and to oligodendroglial curated gene sets for each process regulating oligodendrogenesis (with their positive or negative score attributed for their respective influence in each process). Output for a hub gene or a compound contains a similarity score (correlation score with the

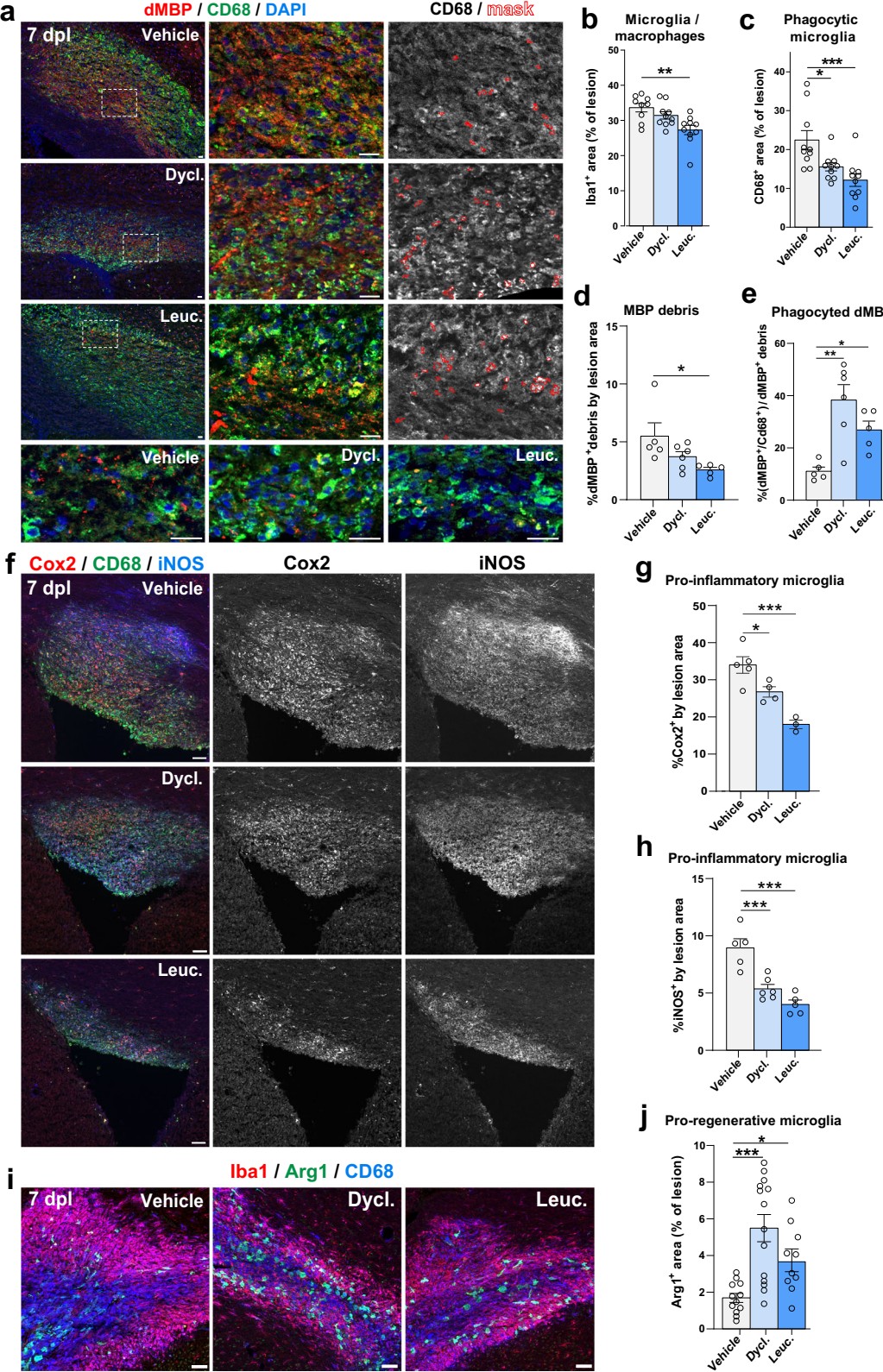

query) and a statistical significance of the similarity score, associated with a probability of finding the same genetic association.

**Pharmacogenomic analysis**

The 156 compounds obtained in common between both approaches were scored using OligoScore and ranked by their scores in the curated target genes for each process of oligodendrogenesis. We also performed a systematic analysis based on gene ontology databases and cell-specific enrichment (see methods for GO analysis) to exclude compounds showing potential side effects (e.g., effects onto other lineages and/or cell death). Finally, we also selected compounds based on their functional properties and signaling pathways modulated using publicly available drug repository websites (DrugBank; KEGG Drug; GeneCards).

**Fig. 9 | Leucovorin and dyclonine accelerate myelin debris clearance and transition from pro-inflammatory to pro-regenerative microglial profiles in a mouse model of adult demyelination. a** Representative images of the lesion territory immunodetection myelin debris with MBP (dMBP, red) and phagocytic microglia with CD68 (green), showing increased dMBP signal inside CD68+ cells in dyclonine- and leucovorin-treated lesions (yellow dots) and *vs.* more dMBP outside CD68+ cells (red dots) in vehicle-treated lesions at 7 days post-lesion (dpl). The right panels illustrate the mask of automatic quantification for CD68 and dMBP colocalization (red labels). The bottom panels are higher magnification images for dMBP dot visualization in different treated lesions. **b**–**e** Barplots and dot plots representing the quantification in the lesion of Iba1+ microglial area (**b**), CD68+ phagocytic area (**c**), myelin debris as a percentage of dMBP area (**d**), and phagocyted myelin debris as the percentage of dMBP area inside CD68+ cells from the total dMBP area (**e**). Note the strong increase of phagocyted myelin debris within dyclonine- and leucovorin-treated lesions. **f** Representative pictures showing immunodetection of microglia/macrophages (CD68+ cells, green) in the lesion area

expressing inflammatory markers (Cox2 in red, and iNOS in blue). **g, h** Quantification of the percentage of lesion area labeled by Cox2 (**g**) and iNOS (**h**) immunofluorescence, in dyclonine-treated (N = 5) and leucovorin-treated (N = 4) mice compared to vehicle (N = 5). Note the decrease in the pro-inflammatory profiles of microglia in leucovorin- and dyclonine-treated conditions. **i** Representative pictures showing immunodetection of microglia/macrophages (Iba1+ cells, red) in the lesion area presenting phagocytic (CD68+ cells, blue) and pro-regenerative (Arg1+ cells, green) profiles. Note the increase in the pro-regenerative profiles of microglia in leucovorin and dyclonine-treated conditions. **j** Histograms representing the density of pro-regenerative microglia, in dyclonine-treated and leucovorin-treated mice compared to vehicle. Data are presented as Mean ± SEM. *p < 0.05; **p < 0.01; ***p < 0.001. Statistics were performed using One-way ANOVA to compare the cell counts across different treatments, followed by Dunnett's test to compare each treatment with Vehicle (control group). Exact p-values, sample sizes (represented in the dot plots), and source data are provided in the Source Data file and in Supplementary Data 2 - Methods Table 3. Scale bars: 20 µm.

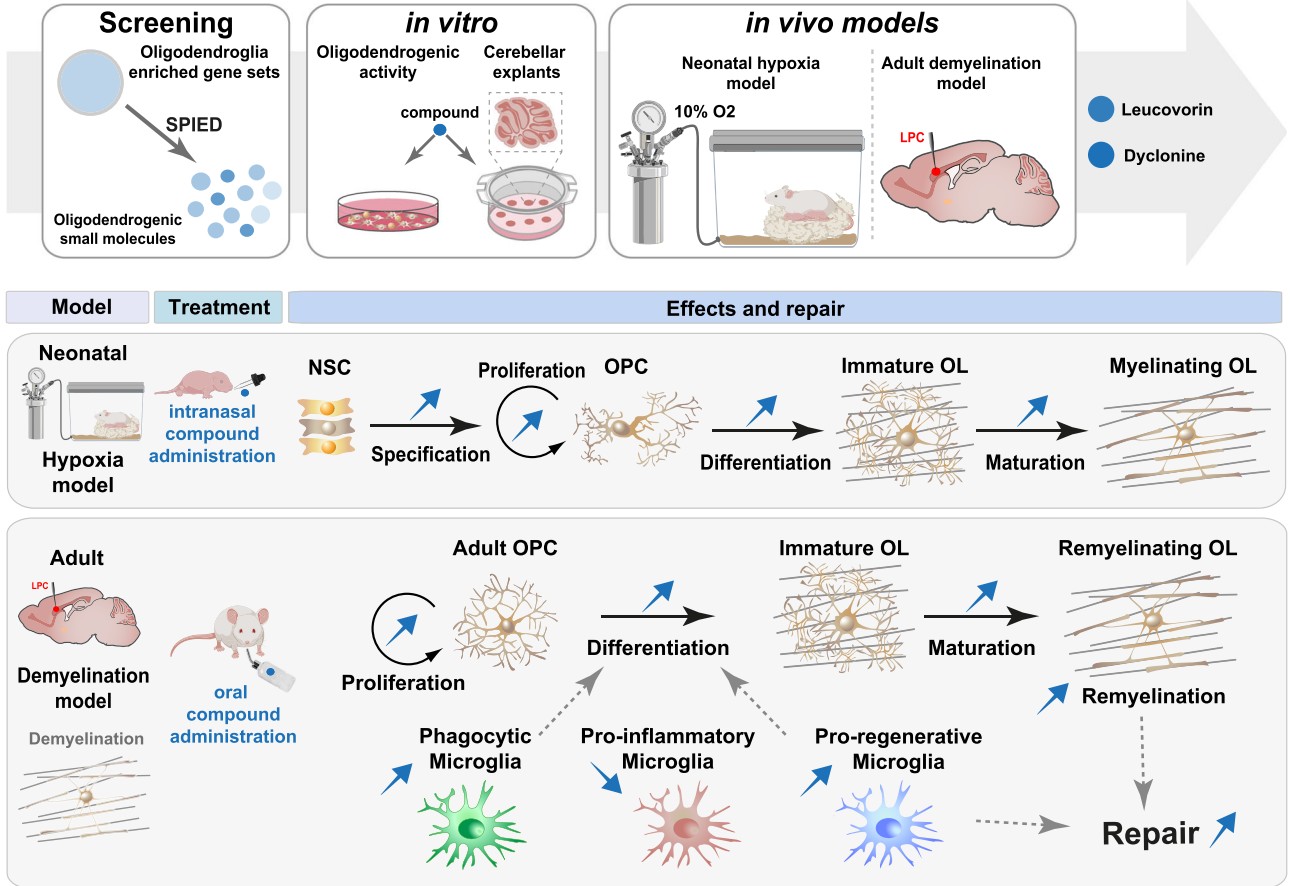

**Fig. 10 | Summary of the study strategy and main findings.** Schematics of the pharmacogenomic approach leading to the identification of small bioactive molecules (compounds) with potential pro-oligodendrogenic activity, followed by the in vitro validation of the top compounds using neural and oligodendrocyte progenitor cell (OPC) cultures as well as organotypic cerebellar explants. The therapeutic efficacy of the top two compounds, leucovorin and dyclonine, both approved by the Food and Drug Administration (FDA), was assessed in vivo using two clinically relevant mouse models of myelin pathologies. In the neonatal

hypoxia mouse model, mimicking some aspects of preterm brain injury, both leucovorin and dyclonine promoted neural stem cell (NSC) differentiation into OPCs and OPC proliferation, with leucovorin additionally restoring the density of myelinating OLs found in normoxic conditions. In an adult focal de/remyelination mouse model of multiple sclerosis, both compounds significantly improved lesion repair in adult mice by promoting OPC differentiation while preserving the pool of OPCs, and by accelerating myelin debris clearance and shifting microglia from pro-inflammatory to pro-regenerative profiles.

## Pharmacological analysis

The ADMET parameters were calculated by the ACD/Percepta software (v.2019.1.2 built 3200) (ACD/Labs Percepta). Predictive models for BBB permeability are based on the rate and extent of brain penetration (respectively log PS and log BB) constants, built using non-linear least squares regression validated by an internal

validation set, and two other experimental external validation sets[109,110]. Physicochemical properties such as lipophilicity (Log P), number of hydrogen bond acceptors and donors, ion form fraction at pH 6.5, and McGowan volume were calculated with the ACD/Labs Algorithm Builder 1.8 development platform and used for the modeling.

The combination of brain/plasma equilibration rate (Log (PS * fraction unbound, brain)) with the partitioning of compounds at equilibrium (Log BB) classifies compounds as either active or inactive on the CNS. The model is validated using experimental data from more than 1500 compounds with CNS activity[111]. This software uses a human expert rules system[112–114] to predict mutagenicity, clastogenicity, carcinogenicity, and reproductive toxicity. The model predicting "health effects on particular organs or organ systems" is based on RTECS and ESIS databases for more than 100.000 compounds. The experimental data have been collected from RTECS and ESIS databases, and diverse publications.

Health effects predictions are known to be very sensitive, so we reduced the weight of these predictions in our selection strategy for the 11 molecules. These results are completed by a reliability index (RI) ranging from 0 to 1, where 0 is unreliable for the prediction, and 1 is a fully reliable prediction. This RI value tends to zero in two cases: first, when the overall similarity between the tested compound and the most similar compounds used for the correction of the global model is weak, and second when an inconsistent variability between the predicted and experimental values was observed among the most similar compounds to the tested compound. Validation sets were used to evaluate result accuracy by computing the Root Mean Square Error (RMSE) between the studied parameter's experimental value and its final predicted value. Results with a reliability index under 0.3 (cut-off value) were discarded. Finally, 11 compounds were selected (Supplementary Data 1 and Supplementary Table 8).

## Animals

We used RjOrl:SWISS mice (Janvier), OF1 mice (Charles Rivers; France), and *Pdgfra-CreERT*; *Rosa26stop-floxed-YFP* mice (RRID:IMSR_JAX:018280; RRID:IMSR_JAX:006148). Both females and males were included in the study. Mice were maintained under standard conditions, with food and water available *ad libitum* in the ICM animal facilities. All animal studies were conducted following protocols approved by local ethical committees and French regulatory authorities (#03860.07, Committee CELYNE APAFIS#187 & 188, Committee DARWIN APAFIS#5516-2016053017288535, APAFIS #38705-2022092718027606).

## Neonatal neural progenitor cultures

NPC proliferating medium: prepared using the following components: DMEM/F12 GlutaMAX™ supplement (life technology, 31331028), 0.6% glucose (Sigma, G8769), 1% penicillin/streptomycin (life technology, 15140122), 5 mM HEPES buffer (life technology, 15630056), 1% N2 supplement (life technology, 17502048), 2% B27 supplement (life technology, 17504044), 20 μg/ml Insulin (Sigma, I6634), 20 ng/ml EGF (Peprotech, AF-100-15), and 10 ng/ml FGF-basic (Peprotech, 100- 18B).

**Preparation and maintenance of NPC cultures.** Pups from RjOrl:SWISS mice (Janvier) at postnatal days 0-1 (P0-P1) were euthanized by decapitation, cortices dissected, and cells mechanically dissociated were collected and washed three times in PBS-1X (Invitrogen, 14080055). Subventricular zones (SVZs) were dissected, transferred to fresh NPC medium, and dissociated using a pipette. Cells were amplified and maintained in a humidified atmosphere at 37 °C and 5% CO2. After 3 days, floating neurospheres were collected by centrifugation at $500 \times g$ for 5 min, dissociated, and resuspended in proliferating NPC medium.

**Plating and treatment.** After two passages, 30.000 cells were plated in 24-well plates with coverslips pre-coated with poly-L-ornithine (sigma, P4957). Compounds were added at concentrations of 250, 500, and 750 nM. Positive controls were added, including 3,3′,5-Triiodo-L-thyronine sodium salt (Sigma, T6397) at 30 mM and clemastine fumarate (Sigma, SML0445) at 500 nM with their respective vehicle (DMSO and

PBS). The medium (including compounds or vehicles) was changed every 2 days.

**Differentiation and fixation.** After 4 days in the NPC proliferating medium, a differentiation medium (without supplemented growth factors) was added for two days. Cells were then washed once in PBS-1X, fixed for 10 min in 4% PBS-paraformaldehyde (Electron Microscopy Sciences, 50-980- 495), and washed three times in PBS-1X.

## Oligodendrocyte precursor cell sorting and cultures

**Brain dissection and cell sorting.** Pups (P4-P7) were euthanized by decapitation, and brains, cortices, and corpora callosa were dissected and dissociated using a neural tissue dissociation kit (P) (Miltenyi Biotec, 130-092-628) with the gentleMACS Octo Dissociator (Miltenyi Biotec, 130-096-427). OPCs were isolated via magnetic cell sorting (MACS) using anti-PDGFRα-coupled-beads (CD140a-PDGFRα MicroBead Kit, Miltenyi Biotec, 130-101-502) and the MultiMACS Cell24 Separator Plus (Miltenyi Biotec, 130-095-691).

**Culture conditions.** 40,000 cells were plated on coverslips pre-coated with poly-L-ornithine (sigma, P4957) and amplified for two/three days in a proliferating medium containing DMEM High Glucose (Dutscher, L0103-500), Ham's Nutrient Mixture F12 (Merck, 51651 C), L-Glutamine (Thermo, 200 mM), Hormone mix (Gritti, A., et al., 2001), Penicillin/ Streptomycin (life technology, 15140122, 1X), 20 ng/ml EGF (Peprotech, AF-100-15), 10 ng/ml FGF-basic (Peprotech, 100-18B), and 10 ng/ ml PDGF-AA (PeproTech, 100-13 A). Differentiation and Treatment: following the proliferation phase, cells were shifted to the differentiation medium for 3 days (without growth factors). Compounds were added to the medium at a concentration of 750 nM. Positive controls, including T3 (3,3′,5-Triiodo- L-thyronine sodium salt, Sigma, T6397) at 30 nM and clemastine fumarate (Sigma, SML0445) at 500 nM were used as positive controls, with their respective vehicles (DMSO and PBS) as negative controls. Cells were fixed for 10 minutes in 4% PBS-paraformaldehyde (Electron Microscopy Sciences, 50-980- 495) and washed three times with PBS-1X.

## Cerebellar organotypic cultures

**Culture preparation.** Pups from newborn (P0) RjOrl:SWISS mice (Janvier) were euthanized by decapitation, and cerebellar organotypic cultures were prepared as previously described[115]. Briefly, cerebellar parasagittal slices (350 μm thick) were cut on a McIlwain tissue chopper and transferred onto 30 mm diameter Millipore culture inserts with 0,4 μm pores (Millicell, Millipore, PIHP03050).

**Culture conditions.** Slices were maintained in incubators at 37 °C, under a humidified atmosphere containing 5% CO2 in six-well plates containing 1 ml of slice culture medium, containing basal Earle's salts medium (BME, sigma, MFCD00217343), 25% Hanks' balanced salt solution (Sigma, H9394), 27 mM glucose (Sigma, G8769), 1% penicillin/ streptomycin (Life technology, 15140122), 1 mM glutamine (Sigma, 228034), and 5% horse serum (New Zealand origin, heat-inactivated; ThermoFisher, 16050122). The medium was renewed every 2 to 7 days to support OPC differentiation and myelination.

**Treatment and fixation.** At this time point, small molecules were added daily for 3 days, including Sm1, Sm2, Sm5, Sm6, Sm7, and Sm11 at 750 nM. Positive controls, including triiodothyronine (T3; Sigma, T6397- 100MG) at 30 nM and clemastine fumarate (SelleckChem, Houston, TX) at 500 nM, were added, with their respective vehicles (DMSO and PBS) as negative controls. After treatment, slices were collected and fixed for 1 h at room temperature in 4% PBS-paraformaldehyde (Electron Microscopy Sciences, 50- 980-495) and incubated for 20 min at 4 °C in Clark's solution (95% ethanol/5% acetic acid), then washed 3 times with PBS 1X.

## Compounds nomenclature

Small molecule (Sm)

| Nomenclature | Compound |
| --- | --- |
| Sm1 | Meticrane (Arresten) |
| Sm2 | Heptaminol |
| Sm3 | melatonin |
| Sm4 | Naringenin |
| Sm5 | Prestwick-674 (Dyclonine HCl) |
| Sm6 | Ginkgolide A |
| Sm7 | Levonorgestrel |
| Sm8 | Medrysone |
| Sm9 | Thioperamide |
| Sm10 | Trihexyphenidyl |
| Sm11 | Calcium folinate |

## Neonatal Hypoxia

**Hypoxic rearing conditions.** P3 OF1 mice were placed for 8 days until P11 in a hypoxic rearing chamber maintained at 10% O2 concentration by displacement with N2 as previously described[116]. A separate group was maintained in a normal atmosphere (normoxic group).

**Compound administration.** Compounds were administered intranasally, with doses corresponding to the oral administration in the adult mice Dyclonine/Sm5 at 5 mg/kg[79,80] and Leucovorin/Sm11 at 0,5 mg/kg in humans[81,82]. Mucus was first permeabilized using type IV hyaluronidase, then 10 μl of compounds (Sigma) were administered 3 times daily from P11 to P13 (starting at the end of the hypoxic period and repeated every 24 h) in sterile PBS (control).

**Proliferation and maturation analysis.** For proliferation analysis, mice were injected with EdU (Sigma) to label cells in the S-phase of the cell cycle. Mice were perfused 1 h post-injection at P13. For analysis of OL maturation, mice were perfused at P19.

**Perfusion and tissue preparation.** Mice were euthanized by IP injection of an overdose of Euthasol (200 μg/g), preceded by an injection of a painkiller (Rompun à 30 μg/g), followed by intracardial perfusion of 4% paraformaldehyde. All perfusions were performed with Ringer's solution, followed by an ice-cold solution of 4% paraformaldehyde (Thermo Fisher). Mice were sacrificed at P13 or P19 by an intraperitoneal overdose of pentobarbital, followed by perfusion with Ringer's lactate solution and 4% paraformaldehyde (PFA; Sigma) dissolved in 0.1 M phosphate buffer (PB; pH 7.4). Brains were removed and postfixed for 24 h at 4 °C in 4% PFA and vibratome-sectioned in 50 μm thick coronal serial free-floating sections.

## Intracerebral demyelination in the adult mouse

**Induction of recombination in adult OPCs.** Tamoxifen (Sigma) was administered orally (gavage) to *Pdgfra-CreERT; Rosa26stop-YFP* mice for five consecutive days (210 g/kg per day, dissolved in corn oil at 20 mg/ml), 10 days before lesion induction to optimize the YFP-labeling of newly- generated OLs in the lesion area.

**Lesion Induction.** Before surgery, 4-month-old RjOrl:SWISS mice (Janvier) and *Pdgfra-CreERT; Rosa26stop-YFP* mice were administered with buprenorphine (30 mg/g) to prevent postsurgical pain. Mice were anesthetized with isoflurane (ISO-VET). Eye protection (Ocrygel, Tvm) and cream lidocaine (Anesderm 5%) were applied to prevent eye

dryness and pain from ear bars. A small incision was made on the head, and liquid lidocaine was applied to the site. Focal demyelinated lesions were induced by injecting 0.5 μl of a 1% lysolecithin solution (L-α-lysophosphatidylcholine, Sigma L4129) diluted in 0.9% NaCl into the corpus callosum. A glass capillary connected to a 10 μl Hamilton syringe was fixed and oriented through stereotaxic apparatus (coordinates: 1 mm lateral, 1.3 mm rostral to Bregma, 1.7 mm deep to brain surface). Animals were left to recover for a few hours in a warm chamber.

**Compound administration.** Dyclonine/Sm5 (5 mg/kg) and Leucovorin/Sm11 (0.5 mg/kg) were administered daily in 5% glucose drinking water according to the reported oral administration in adult mice (dyclonine[79,80];) and humans (leucovorin[81,82];). Given the reported solubility in water and the half-life of these compounds, we diluted them in the drinking water and renewed the treatment daily. Given that a 40 g mouse drinks approximately 6 mL per day, we diluted respectively 0.2 mg of dyclonine/Sm5 (0.2 mg in 40 g = 5 mg/kg) and 0.02 mg of leucovorin/Sm11 (0.02 mg in 40 g = 0.5 mg/kg) in 6 mL, scaling these drug concentrations to a total volume of 200 mL per bottle. We added 5% glucose to the preparation to increase the appetite of the mice.

**Monitoring.** Daily intake volumes were monitored, ensuring no significant differences in consumption between treatments, and confirming the appropriate dosing.

**Perfusion and tissue preparation.** Mice were euthanized by IP injection of an overdose of Euthasol (200 μg/g), preceded by an injection of a painkiller (Rompun à 30 μg/g), followed by intracardial perfusion with 25 mL of 2% PBS-paraformaldehyde freshly prepared from 32% PFA solution (Electron Microscopy Sciences, 50-980-495). Perfused brains were dissected out, and dehydrated in 10% sucrose, followed by 20% sucrose overnight. Brains were then embedded in OCT (BDH), frozen, and sectioned into 14 μm thick sagittal sections using a cryostat microtome (Leica).

## Immunofluorescence staining

**Coated coverslips.** Coverslips were blocked for 30 min at room temperature in a moist chamber using a solution of 0.05% Triton X-100 and 10% normal goat serum (Eurobio, CAECHVOO-OU) in PBS. Coverslips were incubated with primary antibodies for 30 min at room temperature. Following 3 washes with 0.05% Triton X-100/PBS, coverslips were incubated for 30 min at room temperature with secondary antibodies (Molecular Probes or Thermo Fisher) and DAPI (Sigma-Aldrich®, D9542). Coverslips were washed 3 times with 0.05% Triton X-100 in PBS and mounted with Fluoromount-G® (SouthernBiotech, Inc. 15586276).

**Ex vivo cerebellar slices.** Slices were incubated for 20 min at 4 °C in Clark's solution (95% ethanol/5% acetic acid) and washed 3 times with PBS. Then, sections were incubated for 1 h in a blocking solution (0.2% Triton X-100, 4% bovine serum albumin, and 4% donkey serum in PBS). Slices were incubated with primary antibodies for 2 h at room temperature. Slices were washed 3 times with 0.1% Triton X-100 in PBS and incubated with secondary antibodies (Molecular Probes or Thermo Fisher) and DAPI (Sigma-Aldrich®, D9542) in blocking solution at room temperature for 2 h. Slices were then washed 3 times with 0.1% Triton X-100 in PBS and mounted in Fluoromount-G® (SouthernBiotech, Inc. 15586276).

**In vivo cryosections of lesions.** Sections (14-μm thick) of lesions were dried for 30 minutes at room temperature. For MOG staining, sections were treated with 100% ethanol for 10 minutes at room temperature. Sections were permeabilized and blocked in a solution of 0.05% Triton X-100 and 10% normal goat serum in PBS for 1 h. Sections were

incubated overnight with primary antibodies at 4 °C. Sections were washed 3 times with 0.05% Triton X-100 in PBS and incubated with secondary antibodies (Molecular Probes or Thermo Fisher) and DAPI (Sigma-Aldrich®, D9542) during 1 h at room temperature. Sections were washed 3 times with 0.05% Triton X-100 in PBS and mounted with Fluoromount-G® (SouthernBiotech, Inc. 15586276).

**Forebrain sections following neonatal hypoxia.** Free-floating 50 μm coronal sections were collected. Antigen retrieval was performed for 20 min in citrate buffer (pH 6.0) at 80 °C, cooled for 20 min at room temperature, and washed with PBS. Blocking was performed in a TNB buffer (0.1 M PB; 0.05% Casein; 0.25% Bovine Serum Albumin; 0.25% TopBlock) with 0.4% triton-X (TNB-Tx). Sections were incubated with primary antibodies overnight at 4 °C under constant agitation. Following washing in 0.1 M PBS? with 0.4% triton-X (PB-Tx), sections were incubated with secondary antibodies (Jackson or Invitrogen) and DAPI (Life Technologies; D1306) for 2 h at room temperature. The revelation of EdU was done using Click-it, EdU cell proliferation Kit (Thermo Fisher Scientific).

The list of primary and secondary antibodies used can be found in Supplementary Data 2 and Methods Tables 4 and 5.

### Image acquisition and analysis

**Visualization and acquisition.** Cryosections and immunocytochemistry were imaged using Zeiss® Axio Imager M2 microscope with Zeiss® Apotome system at 20X/0.8 NA dry objective (Plan- apochromat), including deconvolution and Z-stack. Acquisition for cerebellar slices was done using a confocal SP8 X white light laser (Leica) at × 40 magnification with a Z-axis range of 10–15 μm, and nine fields (1024 × 1024) were captured and merged into a mosaic.

**Quantifications.** For neonatal hypoxia sections, quantification was performed by automatic quantifications of the Olig2 signal using QuPath software (V0.3.0) in manually defined regions of the corpus callosum and cortex (based on DAPI counterstaining). Other analyses were performed using ZEN (Zeiss) and ImageJ (Fiji) software packages from z-stack images, with macros for ex vivo experiments to determine myelination and differentiation index (methods previously described in ref. 66. Olig2 + cells were detected in five equally spaced coronal sections (50 μm, 250 μm apart). Identification of Olig2 + cells co-expressing GSTπ was performed in manually defined regions covering the entire forebrain (Isocortex, Olfactory areas, Striatum, Fiber tracts, Cortical subplate, Thalamus, Hypothalamus) using the Allen adult mouse brain reference atlas (CCF v3). Manual quantification was done on z-stack confocal images acquired from the hypothalamic lateral zone, using ImageJ-Win-64. In vivo, images of lesions were quantified manually in a double-blinded manner. Illustrations were performed using Illustrator and Adobe Photoshop (Adobe System, Inc.). For electron microscopy analysis, axons were quantified using Fiji software packages.

### Electron microscopy

**Sample preparation.** Mice were euthanized by IP injection of an overdose of Euthasol (200 μg/g), preceded by an injection of a painkiller (Rompun à 30 μg/g), followed by intracardial perfusion with 2.5% glutaraldehyde in 0.1 M sodium cacodylate buffer 0.1 M (Caco buffer) at pH 7.4. Brains were collected and left overnight in the fixative at 4 °C. After rinsing in Caco buffer, brains were sliced into 1mm-thick sections. Sections were selected and rinsed 3 times with Caco buffer and post-fixed with 2% osmium tetroxide Caco buffer for 1 h at room temperature. After an extensive wash (3 × 10 min) with distilled water, they were incubated overnight in 2% aqueous uranyl acetate. Dehydration was done in a graded series of ethanol solutions (2x5min each): 30%, 50%, 70%, 80%, 90%, and 100% (X3). Final dehydration was performed twice in 100% acetone for 20 min. Samples were then progressively infiltrated with an epoxy resin, Epon 812® (EMS, Souffelweyersheim, France), for 2 nights in 50% resin and 50% acetone at 4 °C in an airtight container, then twice for 2 h in pure fresh resin at room temperature. They were embedded in molds and put at 56 °C for 48 h in a dry oven for polymerization of the resin. Blocks were cut with a UC7 ultramicrotome (Leica Microsystems). Semi-thin sections (0.5μm thick) were stained with 1% toluidine blue and 1% borax. Ultra-thin sections (70 nm thick) of the region were recovered on copper grids (200 mesh, EMS, Souffelweyersheim) and contrasted with Reynold's lead citrate (Reynolds, ES (1963)).

**Microscopy.** Ultrathin sections were observed with a Hitachi HT7700 electron microscope (Milexiaé) operating at 100 kV. Images (2048 × 2048 pixels) were taken with an AMT41B camera.

### Statistics and reproducibility

**Software and analysis.** All analyses were performed using GraphPad Prism 6.00 (San Diego, California, www.graphpad.com). For electron microscopy, plot visualizations, and statistics were performed in R.

**Statistics.** The assumption of normality was made for data distribution; formal testing for normality was not performed. Statistical significance was determined using One-Way ANOVA by multiple comparisons to evaluate differences among groups. Dunnett's Post-hoc test was used following one-way ANOVA for pairwise comparisons against a control group, and unpaired $t$ tests were used for comparisons between two independent groups. Data are presented as Mean ± SEM (Standard Error of the Mean). Significance levels were set as: $p$-value < 0.05: * (one asterisk), $p < 0.01$: ** (two asterisks), $p < 0.001$: *** (three asterisks), $p < 0.0001$: **** (four asterisks). Exact p-values, sample sizes (represented in the dot plots), and source data are provided in the Source Data file. A summary of statistical tests used, the number of replicates, and p-values can be found in Supplementary Data 2 - Methods Table 3. Taking into account the 3 R rule of animal experimentation, only in the case of supplemental Fig. 8, we have made a replication of the experiment presented in Fig. 7, with the main purpose of comparing different doses of leucovorin and dyclonine and using clemastine as a pro-oligodendrogenic positive control, as mentioned in the legend of supplementary Fig. 8.

### Reporting summary

Further information on research design is available in the Nature Portfolio Reporting Summary linked to this article.

## Data availability

All data from this study is provided in the main article, Supplementary Data 1 - Supplementary Tables, and source data are provided with this paper. The following datasets were used in the study as detailed in the methods section and in Supplementary Data 1 - Supplementary Table 1 and Supplementary Data 2 - methods table 2: GSE60905, GSE52564, GSE87544, GSE75330, GSE95194, GSE95093, GSE113973, GSE116470.

## Code availability

The code used in this study is available on the GitHub website at https://github.com/ParrasLab/Hure_Parras_2024_paper.

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

## Acknowledgements

This work was supported by funding from the Agence Nationale de la Recherche [ANR NeoRepair (ANR-17-CE16-0009) and ANR NeoReGen (ANR-22-CE17-0029) grants], the French Multiple Sclerosis Foundation (ARSEP-1284), the ICM Carnot projects (#137, #142, and #148), and BDL Capital Management. The research leading to these results has received funding from the national program "Investissements d'avenir" ANR-10-IAIHU-0006 and the national program "Investissements d'avenir" ANR-11- INBS-0011 – NeurATRIS: Translational Research Infrastructure for Biotherapies in Neurosciences. O.R. lab is supported by the Fondation pour la Recherche Médicale (FRM: EQU202203014658) as well as by the LABEX CORTEX (ANR-11- LABX-0042) of Université de Lyon, within the program "Investissements d'Avenir" (decision n° 2019-ANR-LABX-02) operated by the French National Research Agency (ANR). We thank Severine Candelier and Mathilde Lannes for creating the OligoScore website, and François Xavier Lejeune for overseeing the statistical analysis. We thank the following ICM (Paris Brain Institute) core facilities: PHENOPARC, CELIS, Histomics, and ICM.QUANT, and all personnel involved for their contribution and support.

## Author contributions

Conceptualization, supervision, funding acquisition: C.P. and O.R. Design of the experiments: C.P., O.R., and J.B.H. Achievement of the experiments and data analysis: bio-informatics, J.B.H., C.P., O.R., and R.A.; culture models, J.B.H., L.M.G, C.M., L.B., and A.A.N.; hypoxia model, L.B.O., L.F., N.V., and O.R.; adult demyelination model, J.B.H., L.M.G, C.P., C.M., N.I., A.A.N., and M.P.; pharmacology, J.A.C., M.A.D., R.T., and F.G. Writing of the original version of the manuscript: C.P. with main contributions from J.B.H., O.R., B.A.H., and help from all other authors.

## Competing interests
Jean-Baptiste Hure, Carlos Parras, Olivier Raineateau, and Louis Foucault are inventors in the patent entitled 'Organic molecules for 1312 treating myelin pathologies', published on September 28th, 2023 (WO2023180474) that is based on most of the data published in this study. Patent applicants: ICM - Paris Brain Institute, INSERM, CNRS, AAPHP, Sorbonne Université, Université Claude Bernard Lyon 1. The other authors declare no competing interests.
