## [Transparent Peer Review file · Nature Communications]

Pharmacogenomic screening identifies and repurposes leucovorin and dyclonine as pro-oligodendrogenic compounds in brain repair

Corresponding Author: Dr Carlos Parras

This manuscript has been previously reviewed at another journal. This document only contains reviewer comments, rebuttal and decision letters for versions considered at Nature Communications.

Version 0:

Reviewer comments:

Reviewer #1

(Remarks to the Author)

This manuscript aimed to find new therapeutic solution for some very challenging diseases, namely, early preterm-birth brain injury and adult multiple sclerosis. This is very clinically important. The authors creatively proposed a very novel strategy to identify the drug targets for these disease by the full utilization of genetic data and bioinformatics. These efforts indeed generate some potential lead compounds and their effect were subsequently proved by both in vitro and in vivo experiments. Generally, this study has very good creativity which was support by solid experiment data. The findings of this manuscript could also provide useful hints for the development of new medication of early preterm-birth brain injury and adult multiple sclerosis. Whereas the quality of this manuscript could be further improved in the following aspects.

1) Line 153, the authors developed a very novel scoring system useful for the identification of effective compounds using the genetic data in this study. I am wondering whether this scoring system has been validated somewhere else before it was employed in this study.

2) Line 177, the authors ranked the potential of compounds as a lead compound by evaluate various properties of druggability such as toxicology, pharmacokinetics and pharmacodynamics. This is a correct and smart strategy. After checking Supplementary table 8, we can see that the authors have paid too much attention on the issue of toxicology. Most of evaluation items in Supplementary table 8 are related to toxicology rather than pharmacokinetics and pharmacodynamics. In my opinion, the identification of suitable candidates for the proof of concept should focus on their pharmacokinetics and pharmacodynamics characteristics. However, I found very few items of scoring related to pharmacokinetics and pharmacodynamics characteristics. Even an important scoring item related to pharmacokinetic of CNS drug, namely, BBB penetration is just a very simple qualitative prediction provided by the software of ACDsee. Actually, many of the compounds in Supplementary table 8 are old drugs. I guess it is possible to collect the BBB penetration data in animals from the literature. Such quantitative information may be more valuable for such scoring purpose.

3) Line 621, please specify the names of the compounds here. Please also clarify the compounds information in other parts of this manuscript such as line 300.

4) Line 637, in the animal experiment, the authors used a dose of 30 mg/g buprenorphine. This is a very high dose considering the human dose is usually less than 1 mg. How can the mice tolerate such a high dose in the experiment? Additionally, the oral bioavailability of buprenorphine is low due to the extensive first pass effect. Thus this drug is administered by injection or sublingual administration. The authors didn't describe the administration method. This should be clearly described.

5) Line 645-647, why did the authors choose to administer the drug via the drinking water? Because it is difficult to control the exact dose in this way. What is the actual dose for mice in this study? I can't get the accurate information from the description like this: (sm5 at 5 mg/kg and 647 sm11 at 0,5 mg/kg).

6) It is important to set different dose levels of compounds for both in vitro and in vivo studies to demonstrate the dose or exposure/efficacy relationship to some extent. I am not sure if the authors could improve the quality of the manuscript in this aspect.

Reviewer #2

(Remarks to the Author)

Huré et al. present a pharmacogenomic approach to identify small bioactive molecules with pro-oligodendrogenic activity. Using a curation strategy (OligoScore), they identify a series of small bioactive molecules with pro-oligodendrogenic activity, based on their effect on transcriptional pathways linked to oligodendrogenesis and (re)myelination. They demonstrate that these compounds are able to induce NPC differentiation to oligodendroglial fate, boost oligodendrocyte differentiation, and promote myelination. They then selected Sm5 (dyclonine) and Sm11 (leucovorin) as candidates to move forward in preclinical models of myelin pathologies. They found that both were able to promote oligodendroglial regeneration in a mouse model of preterm brain injury and enhance oligodendrogenesis in the LPC model of de/remyelination in the corpus callosum.

Major Points

1. While the authors do a nice job showing the effect of the compounds on several metrics in various models, the claim that the compounds promote myelination needs to be strengthened. The only evidence provided is the myelin index in cerebellar slice cultures.
 - a. For the in vivo models, the authors show effects on a variety of OL lineage markers. However, it would be important to look at myelination (by IHC and EM, for instance) in both the preterm brain injury model and the LPC model. For instance, reports have shown hypomyelination in the preterm brain injury model (e.g. Yuen TJ, Silbereis JC et al., Cell 2014), so an increase in myelination here with compound treatment would be interesting. In addition, in Figure 7, the title says that compound treatment promotes myelin repair, but there is no data on this to corroborate this claim.
 - b. The in vivo evidence of compounds boosting OL differentiation and maturation could be stronger. For instance, Fig 6I/J show that Hx + compound is not significantly different from Nx, but only leucovorin treatment is significantly boosting OL differentiation / maturation over the Hx condition, which seems to be the more relevant comparison here.
 - c. Figure 4 – it would be helpful to also look at number of Plp+ (or another mature OL marker, like MBP) to see if these compounds affect numbers of differentiated cells, versus an effect on morphology, which the authors report.
2. The compound dosing paradigm for in vivo studies need corroboration. For instance, in the LPC studies, compounds were dosed in the drinking water, so there is no controlled way to monitor the amount of drug being delivered. Were any measures of PK and/or target engagement done? This would help strengthen things. Could the authors try PO dosing?
3. The link to microglia, particularly the change in phagocytic / pro-regenerative microglial numbers, is intriguing but also needs to be strengthened. Did the authors look at the presence of myelin debris (e.g. dMBP staining)? Were changes in inflammatory signals observed? In Fig 7O, it also appears that lesion size might vary. Was this examined?
4. The bioinformatics data analysis needs clarification:
 - a. The authors mention that raw sequencing data was downloaded from GEO, but it appears that the authors applied a microarray data processing pipeline. Can this be clarified?
 - b. In addition, no details were documented about how the authors applied Seurat for single-cell RNAseq data analysis and for bulk RNAseq analysis. For instance, how did the authors correct for different batches for data integration? How many cells were integrated? What statistical method was used for differential analysis? Did the authors use adjusted p-value or a raw p-value as cutoff?

Minor Points

1. The sentence in the abstract “An unmet challenge is the discovery of medications presenting convincing pro-oligodendrogenic activity for the eventual clinical treatment of these pathologies” is somewhat misleading as there are several compounds currently being tested (e.g. clemastine, analogs of thyroid hormone) that promote OPC differentiation and remyelination. Of note, this is nicely discussed in the introduction.
2. Figure 3 – Why was 750nM chosen as the dose to test all compounds (also 250 and 500nM in Figure 3, extended data 1)? Did you do a wider dose response (e.g. higher doses)?
3. Figure 4 – CaBP staining is in green, but the CaBP+ axons are shown in white?
4. For the slice cultures, it would be helpful to have another metric of myelination, perhaps Caspr staining normalized to axons, which would mark compact myelin and not be confounded by MBP+ oligodendrocytes on axons. It is intriguing that clemastine promotes OL differentiation, but not myelination, while T3 does not promote differentiation, but promotes myelination.
5. Discussion on why Sm2 was not selected to move forward to in vivo studies would be helpful. Is it simply that it was ranked #3 (as shown in Figure 5, extended data 1), or was there additional rationale?
6. The manuscript could benefit from proofreading throughout (some grammar errors and typos).

Reviewer #3

(Remarks to the Author)

This manuscript tackles an important aspect of debilitating diseases such as newborn brain injury and multiple sclerosis, that of identifying therapeutic agents which promote remyelination in injury. Several other studies have used cell screening approaches in vitro and in vivo to identify compounds which promote remyelination. The novelty of this paper lies in the demonstration of an informatics approach to predict compounds which will induce oligodendrogenesis. The authors use this approach to identify predicted compounds which will promote remyelination after injury, and use two of them in vivo to promote injury repair. I like the idea of this novel informatics approach a lot, but have quite a few issues with its execution:

- 1) It seems unclear from the description how SPIED is predicting bioactive molecules which induce transcriptional changes that induce oligodendrogenesis. A more thorough description of how this tool is used would be useful.

- 2) The main novelty of this paper is the informatics approach used. The authors discuss different gene sets for different processes of oligodendrocyte biology, specification, proliferation, migration, survival, differentiation, myelination. But it seems as if they lump all processes together to identify drugs that are 'oligodendrogenic'? Were compounds assessed that effect specific individual processes?
- 3) A somewhat similar point, of the compounds selected, which transcriptional hubs are activated by each and which individual processes are predicted to be affected by this?
- 4) In both in vivo hypoxia and lysolecithin models, the chosen compounds are described as increasing OPC proliferation as well as the number of differentiated OL. It is difficult to understand how a drug is affecting both of these two very different processes. It is important to describe gene hubs affected by the drugs and how this leads to effects on proliferation and differentiation.
- 5) The most confusing part of the manuscript comes at the very end. The point of the paper is to identify compounds which affect oligodendrocyte transcriptional networks, but then at the very end the authors seem to be suggesting that the compounds they chose also alter a microglial signature in vivo in injury, raising the possibility that the promotion in remyelination has nothing to do with the effects on OPCs but due to altered microglia.

Version 1:

Reviewer comments:

Reviewer #2

(Remarks to the Author)

Thank you to the authors for their revision and additional experiments.

While the authors conducted numerous additional experiments and useful discussion, which help strengthen their claims, there are still points that are central to their claims that need to be strengthened.

- 1) The authors main aim is to identify compounds that promote repair that can translate to pathologies such as early preterm-birth brain injury (PBI) and adult multiple sclerosis (MS). They have done extensive work to show the enhancement of proliferation and oligodendroglial fate, as well as differentiation. However, as the ultimate aim is to promote myelination or remyelination, this still needs to be strengthened.
 - a. The fate mapping experiments (Figure 8) add to the story, but still do not address the myelination component. For instance, there could be many mature oligodendrocytes, which does not translate into myelin formation. The claim that these are (re)myelinating oligodendrocytes is not supported with data.
 - b. Did the authors examine a quantitative or semi-quantitative measure of remyelination in their EM study? G-ratios likely are not a very useful metric in the corpus callosum.
 - c. Figure 6f, h, j – was the difference between Hx vs. Hx + Dycl. statistically significant? It is not shown, but the text is inconsistent in the results (mentioning a rescue and then more mitigated effects).
 - d. The authors conclude that the compounds have the capacity to cross the BBB, but no data are shown to directly measure compound concentration in the brain to support this conclusion?
- 2) Thank you for adding Methods Table 2 to summarize the expression datasets included in this study and the statistics for selecting DEGs. However, the single-cell data processing documented in lines 597-600 is still confusing. How did the authors apply both 'min genes=500 and max genes<200' filtering conditions simultaneously? Why are the authors interested in cells with high mitochondrial percentage (>10%)?
 - a. The following link may help with general guidance: https://satijalab.org/seurat/articles/pbmc3k_tutorial.html
 - b. I assume the authors used the first 20 PC dimensions for UMAP or tSNE clustering with a clustering reduction of 0.9? For the 1,598 DEGs identified, which statistical method was used here?

Reviewer #3

(Remarks to the Author)

I appreciate the authors responses to my questions, and feel now they have addressed all the concerns.

Version 2:

Reviewer comments:

Reviewer #2

(Remarks to the Author)

I greatly appreciate the diligence and thoroughness that the authors have put forward in addressing my comments. I feel that they have now sufficiently addressed all my concerns.

Point by point response to reviewers' comments

Answers in blue.

Reviewers' comments:

Reviewer #1 (Remarks to the Author):

This manuscript aimed to find new therapeutic solution for some very challenging diseases, namely, early preterm-birth brain injury and adult multiple sclerosis. **This is very clinically important.** The authors **creatively proposed a very novel strategy** to identify the drug targets for these diseases by the **full utilization of genetic data and bioinformatics.** These efforts indeed generate some potential lead compounds, and their effect were subsequently proved by both in vitro and in vivo experiments. Generally, this study has **very good creativity** which was supported by **solid experiment data.** The findings of this manuscript **could also provide useful hints for the development of new medication of early preterm-birth brain injury and adult multiple sclerosis.** Whereas the quality of this manuscript could be further improved in the following aspects.

We thank the reviewer for her/his enthusiastic comments concerning the clinical relevance, creativity, and solidity of the experimental data of our study. We answer to the point raised below.

1) Line 153, the authors developed a very novel scoring system useful for the identification of effective compounds using the genetic data in this study. I am wondering whether this scoring system has been validated somewhere else before it was employed in this study.

To answer the reviewer: (1) in the context of a recent collaboration (Nat. Commun., DOI: 10.1038/s41467-023-36846-w), we used OligoScore to analyze bulk RNA-seq data from spinal cord of EAE mice subjected to different treatments, not only confirming at the transcriptional level the deregulated processes of oligodendrogenesis previously found by immunofluorescent analysis, but also identifying key genes responsible for these deregulations; (2) in our revised manuscript, we added two further validations. First, we queried OligoScore with genes enriched in OPCs and myelinating OLs (mOLs), using transcriptomes from purified brain cells (Zhang et al., 2014, RNAseq) or from single cells (Marques et al., 2016, 2018). With both datasets, as expected, we found that genes enriched in OPCs are involved in several processes of oligodendrogenesis (specification, promoting proliferation, differentiation, and (re)myelination), while genes enriched in mOLs are mainly promoting differentiation and myelination (Figure 2 extended data 2a-d). Second, we queried OligoScore with genes deregulated in OPCs upon either a genetic (*Chd7*-deletion in OPCs; Marie et al., PNAS 2018) or environmental perturbation (IL1b-mediated neonatal neuroinflammatory model, Schang et al., 2022). Remarkably, OligoScore analysis not only highlighted the deregulated processes of oligodendrogenesis previously found by immunofluorescent analyses in these studies (that is, reduced survival and differentiation of *Chd7*-iKO OPCs, and reduced OPC differentiation paralleled by an increase proliferation of OPCs from IL1b-induced neuroinflammatory model), but also provided mechanistic insights identifying some of the genes responsible for the deregulation of each process. We have added these analyses to the manuscript as follows: (a) results section lines 165-170 and Figure 2 Extended Data 2, (b) gene sets used to query and validate OligoScore in the Excel file named 'OligoScore validation tables', (c) a method's section named 'Validation of OligoScore strategy' describing these validations.

2) Line 177, the authors ranked the potential of compounds as a lead compound by evaluate various properties of druggability such as toxicology, pharmacokinetics and pharmacodynamics. This is a correct and smart strategy. After checking Supplementary table 8 , we can see that the authors have paid too much attention on the issue of toxicology. Most of evaluation items in Supplementary table 8 are related to toxicology rather than pharmacokinetics and pharmacodynamics. In my opinion, the identification of suitable candidates for the proof of concept should focus on their pharmacokinetics and pharmacodynamics characteristics. However, I found very few items of scoring related to pharmacokinetics and pharmacodynamics characteristics. Even an important scoring item related to pharmacokinetic of CNS drug, namely, BBB penetration is just a very simple qualitative prediction provided by the software of ACDsee. Actually, many of the compounds in Supplementary table 8 are old drugs. I guess it is possible to collect the BBB penetration data in animals from the literature. Such quantitative information may be more valuable for such scoring purpose.

According to the reviewer's suggestion, we searched the literature for those compounds experimentally tested for their BBB penetration and found data supporting most of the predictions (83%, 23 out of 28 molecules). For the 10 molecules with no experimental data, we have now labeled them as '(Predicted)'. We included this data and the bibliographic references for the 28 molecules with available experimental values in the supplemental table S8 columns D and E, and provide the details in the Word file named *Reference S8 BBB*.

In more general aspects, we selected compounds based on: (1) their larger pro-oligodendrogenic activities (high pharmacogenomic total scores), (2) their crossing of the BBB, and (3) not having major predicted toxicity. Finally, for the *in vivo* studies, we selected the best two being FDA-approved, so that their pharmacokinetics and pharmacodynamics did not need to be repeated. We have modified the main text to make this point clear in **lines 180-186**.

3) Line 621, please specify the names of the compounds here. Please also clarify the compounds information in other parts of this manuscript such as line 300.

As suggested by the reviewer in our revised manuscript we have added a Method section called 'Compounds nomenclature' and added the dyclonine (Sm5) and leucovorin (Sm11) in the text of the *in vivo* studies.

4) Line 637 , in the animal experiment, the authors used a dose of 30 mg/g buprenorphine. This is a very high dose considering the human dose is usually less than 1 mg. How can the mice tolerate such a high dose in the experiment? Additionally, the oral bioavailability of buprenorphine is low due to the extensive first pass effect. Thus this drug is administered by injection or sublingual administration. The authors didn't describe the administration method. This should be clearly described.

We thank the reviewer for spotting this error in the methods which we have corrected. Indeed, buprenorphine was administered by intraperitoneal injection at 0.1 µg/g to prevent postsurgical pain, in agreement with our approved "animal experimentation procedure" (APAFIS #38705-2022092718027606 v3).

5) Line 645-647, why did the authors choose to administer the drug via the drinking water? Because it is difficult to control the exact dose in this way. What is the actual dose for mice in this study? I can't get the accurate information from the description like this: (sm5 at 5 mg/kg and sm11 at 0,5 mg/kg).

Oral administration is a common option for administering leucovorin (aka calcium folinate/Sm11) in humans and is often well tolerated by patients, and dyclonine hydrochloride (Sm5) has been used as a local oral anesthetic in humans for more than 50 years. We privileged the oral administration via the bottle of drinking water to post-lesion adult mice to **(1) prevent stress induced inflammation as a confounding parameter** (scientific rational), and **(2) avoid unnecessary stress/suffering to post-operative animals** as demanded by the ethical guidelines approved by our institution (ethical rational). Importantly, as we indicated in the Fig. 7 Extended Data 1, we monitored daily the intake volume, finding no differences in the amount of drinking intake between the different treatments, thus confirming the daily oral posology of 5mg/kg of dyclonine/Sm5 and 0.5mg/kg of leucovorin/Sm11, per mouse per day.

The previously reported dose and frequency of oral administration for leucovorin and dyclonine are variable between studies. The typical concentration of leucovorin used in humans ranges between 0.5 to 2.5 mg/kg per day. However, other studies have demonstrated the compound's effectiveness at higher doses (2 to 10 mg/kg per day) in the context of ASD patients (Frye et al., 2018; Rossignol and Frye, 2021). We thus chose to administer leucovorin at a concentration of 0.5mg/kg per day over 7 days. For dyclonine, with two studies administering adult mice in the range of 1 to 10 mg/kg (Sahdeo et al; 2014) or at 5 mg/kg (Okazaki et al; 2018), we chose to administer the mean dose of 5 mg/kg. We used the following rationale for the treatment: considering the reported solubility in water and half-life of these compounds, we diluted them in the drinking water and renewed the treatment daily. Given that a 40 g mice drinks approximately 6mL per day, we diluted respectively 0.2mg of dyclonine/Sm5 (0.2mg in 40 g = 5 mg/kg) and 0.02mg of leucovorin/Sm11 (0.02mg in 40 g = 0.5 mg/kg) in 6mL. We then scale these concentrations to a total volume of 200mL per bottle. We added 5% glucose to the preparation to increase appetite for the mice. According to the reviewer comment, we have rephrased this part of the material and methods in **lines 817-828**.

6) It is important to set different dose levels of compounds for both *in vitro* and *in vivo* studies to demonstrate the dose or exposure/efficacy relationship to some extent. I am not sure if the authors could improve the quality of the manuscript in this aspect.

In line with the reviewer comment, we used NPC cultures to perform a systematic testing the pro-oligodendrogenic activity **dose effect *in vitro*** of the **11 selected compounds at 3 different concentrations** (250 nM, 500 nM, and 750 nM) based on the previous reports (Mei et al., 2014; Najm et al., 2015; Eleuteri et al., 2017) and in **4 replicated experiments** (Figure 3 Extended data 1). Whole coverslip automatic quantifications of Sox10+ cells indicated that all selected compounds presented the strongest effects to promote Sox10+ oligodendroglia at 750nM. Given the results obtained in this extensive *in vitro*, we carry out following *in vitro* and *ex vivo* experiments at this optimal dose (750 nM). We have added a paragraph in the main text of make this point clearer **in lines 191-198**.

Concerning the *in vivo* compounds' treatments showing robust pro-oligodendrogenic effects in two pathological models, the concentrations used were based on previously reported *in vivo* administration to mice and humans (Sahdeo et al; 2014; Okazaki et al; 2018; Frye et al., 2018; Rossignol et al., 2021).

In this context and considering the 3R ethical rule guidelines for animal experimentation, it is worth considering that in our first version of the study, we already used ~150 adult and ~60 postnatal animals. Nevertheless, taking into account that these experiments are laborious due to the complexity of both models of oligodendrocyte pathology, **in our revised study we addressed both the dose and the mode of administration** (see reviewer 2 point 2.3), performing a new experiment

with 30 adult animals, and also using the pro-remyelinating drug **clemastine** to compare with our compounds. Considering that the doses used before were in the lower range of those previously reported (Sahdeo et al; 2014; Okazaki et al; 2018; Frye et al., 2018; Rossignol et al., 2021), **we tried oral administration by gavage** (theoretically better controlling the dose administered) at two concentrations, the same concentration (to compare with the drinking water treatment) **and a higher dose**. This corresponded to 2X-dose for dyclonine/Sm5 (10mg/kg to be at the highest range used in adult mice) and 10X-dose of leucovorin/Sm11 (5mg/kg, to be in the higher range used in humans).

After two days of gavage, we found clear signs of stress in all groups, including weight loss. Therefore, from the third day onward, we came back to the posology through the drinking water, with mice showing no more signs of stress and weight loss. Immunofluorescent analysis at 7 days post-lesion, to compare with our previous experiment, showed similar pro-oligodendrogenic effects at both doses tested (Fig. 7 extended data 1d-h), and confirmed our previous experiment (Fig. 7f-i). Moreover, while clemastine induced an increase in iOL2 density similar to dyclonine and leucovorin (Fig. 7 extended data 1h), **both leucovorin and dyclonine increased the density of proliferating OPCs** (Fig. 7m, o), **an effect that was absent for clemastine** (Fig. 7o). We have added this data in **lines 352-361**.

In the context of the hypoxia model, the intranasal administration of these compounds immediately following hypoxia is a strategy representing a noninvasive technique for drug administration, that was already demonstrated to impact brain cell behavior and eventually their proliferation (Scafidi et., 2013; Azim et al., 2017). With this treatment, we showed robust increase in the number of proliferating OPCs, progenitors fated to oligodendroglia, and an increase in the number of myelinating OLs in the leucovorin/Sm11 treatment (Fig. 6). Of course, this is without any intention to use intranasal administration in human preterm babies, which receive intravenous administration for most treatments. Our follow up project, NeoReGen, utilizes different mouse models of preterm brain injury to assess in more details the translational capacity of leucovorin and dyclonine to foster brain repair in the context of early preterm brain injury.

Reviewer #2 (Remarks to the Author):

Huré et al. present a pharmacogenomic approach to identify small bioactive molecules with pro-oligodendrogenic activity. Using a curation strategy (OligoScore), they identified a series of small bioactive molecules with pro-oligodendrogenic activity, based on their effect on transcriptional pathways linked to oligodendrogenesis and (re)myelination. They demonstrate that these compounds are able to induce NPC differentiation to oligodendroglial fate, boost oligodendrocyte differentiation, and promote myelination. They then selected Sm5 (dyclonine) and Sm11 (leucovorin) as candidates to move forward in preclinical models of myelin pathologies. They found that both were able to promote oligodendroglial regeneration in a mouse model of preterm brain injury and enhance oligodendrogenesis in the LPC model of de/remyelination in the corpus callosum.

Major Points

1. While the authors do a nice job showing the effect of the compounds on several metrics in various models, the claim that the compounds **promote myelination needs to be strengthened**. The only evidence provided is the myelin index in cerebellar slice cultures.

1.a. For the in vivo models, the authors show effects on a variety of OL lineage markers. However, it would be important to look at myelination (by IHC and EM, for instance) in both the preterm brain injury model and the LPC model. For instance, reports have shown hypomyelination in the preterm

brain injury model (e.g. Yuen TJ, Silbereis JC et al., Cell 2014), so an increase in myelination here with compound treatment would be interesting. In addition, in Figure 7, the title says that compound treatment promotes myelin repair, but there is no data on this to corroborate this claim.

We thank the reviewer for the comments. It is worth mentioning that we do not claim that our compounds directly increase the process of myelin formation but we now provide further evidence showing that they boost the generation of newly-formed OLs and myelinating OLs [expressing GST π , a marker restricted to myelinating OLs (Zhou et al., 2021)] *in vivo* in two models of oligodendrocyte/myelin pathology.

According to the reviewer's comment, to strengthen this aspect of the study, **we performed a new experiment using 30 additional adult mice genetically labeling and quantifying the newly-formed OLs in the lesion area.** To this aim, we fluorescently labeled adult OPCs and their progeny (OPCs and OLs) by administering tamoxifen to *Pdgfra-CreER^T; Rosa26^{stop-YFP}* mice just before inducing the LPC lesion. At 10 days post-lesion, when newly-formed (re)myelinating OLs are found in the lesion area, the quantification of the number of YFP⁺ (newly-formed) oligodendroglia expressing GST π , a marker restricted to myelinating OLs (Zhou et al., 2021), showed that dyclonine and leucovorin increased more than 2-fold the number of newly-formed myelinating OLs (YFP⁺/GST π ⁺ cells) in the lesion area, similar to clemastine (Fig. 8a-d). We have added this data in the result section lines 365-373.

Moreover, balancing the reviewer comment and the 3R ethical guidelines, we have also performed ultrastructural analyses of the effect of our compounds in (re)myelination in the adult model of myelin pathology, using 20 adult mice, including a group treated with clemastine as positive control. At 10 days post-lesion, we confirmed by ultrastructural analysis the increase in myelinating OLs (identified by their round- or oval-shape nucleus having densely packed chromatin and processes wrapping around myelinated axons) in animals treated with leucovorin, dyclonine, and clemastine compared to vehicle-treated animals (Fig. 8e). Moreover, quantification of myelinated axons in the lesion area showed a tendency to decrease the g-ratio of axons in leucovorin-, dyclonine-, and clemastine-treated animals compared to vehicle-treated controls, suggesting an increased remyelination (more wrapping) induced by the compound treatment, that reach significance in the case of leucovorin treatment on axons larger than 1 μ m (Fig. 8f-h). We have added this in lines 372-380.

1.b. The *in vivo* evidence of compounds boosting OL differentiation and maturation could be stronger. For instance, Fig 6I/J show that Hx + compound is not significantly different from Nx, but only leucovorin treatment is significantly boosting OL differentiation / maturation over the Hx condition, which seems to be the more relevant comparison here.

To answer the reviewer point, in the context of the **neonatal hypoxia model**, we now **provided more complete analyses**, increasing the number of animals, and performing automatic and manual quantifications in different brain regions, now provided in Figure 6 and Fig.6 extended data 1f. and detailed in lines 290-320.

1.c. Figure 4 – it would be helpful to also look at number of Plp⁺ (or another mature OL marker, like MBP) to see if these compounds affect numbers of differentiated cells, versus an effect on morphology, which the authors report.

According to the reviewer comment, we now present automatic quantifications of total cells in OPC differentiation cultures after three days of compound's treatment (Figure 4 extended data 1), showing that, while no significant changes were seen in the variable density of PDGFRA+ OPCs (Fig. 4b, c), except for Sm7, both T3, clemastine, and all tested compounds (Sm1, Sm2, Sm5, Sm6, and Sm11) increased the number of MBP+ OLs compared to vehicle-treated controls (Fig. 4b, d). These results demonstrate the capacity of most selected compounds to cell-autonomously foster OPC differentiation. We have included this in **lines 226-231**.

2.1 The compound dosing paradigm for in vivo studies need corroboration. For instance, in the LPC studies, compounds were dosed in the drinking water, so there is no controlled way to monitor the amount of drug being delivered.

Oral administration is a common option for administering leucovorin (aka calcium folinate/Sm11) in humans and is often well tolerated by patients, and dyclonine hydrochloride (Sm5) has been used as a local oral anesthetic in humans for more than 50 years. We privileged the oral administration via the bottle of drinking water to post-lesion adult mice to **(1) prevent stress induced inflammation as a confounding parameter** (scientific rational), **and (2) avoid unnecessary stress/suffering to post-operative animals** as demanded by the ethical guidelines approved by our institution (ethical rational). **Importantly, as we indicated in the Fig. 7 Extended Data 1, we monitored daily the intake volume, finding no differences in the amount of drinking intake between the different treatments, thus confirming the daily oral posology of 5mg/kg of dyclonine/Sm5 and 0.5mg/kg of leucovorin/Sm11, per mouse per day.**

The previously reported dose and frequency of oral administration for leucovorin and dyclonine are variable between studies. The typical concentration of leucovorin used in humans ranges between 0.5 to 2.5 mg/kg per day. However, other studies have demonstrated the compound's effectiveness at higher doses (2 to 10 mg/kg per day) in the context of ASD patients (Frye et al., 2018; Rossignol and Frye, 2021). We thus chose to administer leucovorin at a concentration of 0.5mg/kg per day over 7 days. For dyclonine, with two studies administering adult mice in the range of 1 to 10 mg/kg (Sahdeo et al; 2014) or at 5 mg/kg (Okazaki et al; 2018), we chose to administer the mean dose of 5 mg/kg. We used the following rationale for the treatment: considering the reported solubility in water and half-life of these compounds, we diluted them in the drinking water and renewed the treatment daily. Given that a 40 g mice drinks approximately 6mL per day, we diluted respectively 0.2mg of dyclonine/Sm5 (0.2mg in 40 g = 5 mg/kg) and 0.02mg of leucovorin/Sm11 (0.02mg in 40 g = 0.5 mg/kg) in 6mL. We then scale these concentrations to a total volume of 200mL per bottle. We added 5% glucose to the preparation to increase appetite for the mice. According to the reviewer comment, we have rephrased this part of the material and methods in **lines 817-828**.

2.2 Were any measures of PK and/or target engagement done?

As mentioned in the introduction, results, and discussion, our drugs/compounds have been selected by their large pro-oligodendrogenic activity involving different cellular (neural progenitor/stem cells, OPCs, and microglia) and molecular target mechanisms (including the one carbon/ folic acid metabolism in the case of leucovorin/sm11 and NRF2 antioxidant activities in the case of dyclonine/sm5). Furthermore, for the *in vivo* models of myelin pathologies, we selected FDA-approved compounds previously used in human and animal models. Therefore, it is clearly beyond

the scope of this large and extensive study to perform further detailed measures of pharmacokinetics in animal models.

2.3 This would help strengthen things. Could the authors try PO dosing?

Concerning the *in vivo* compounds' treatments showing robust pro-oligodendrogenic effects in two pathological models, the concentrations used were based on previously reported *in vivo* administration to mice and humans (Sahdeo et al; 2014; Okazaki et al; 2018; Frye et al., 2018; Rossignol et al., 2021).

In this context and considering the 3R ethical rule guidelines for animal experimentation, it is worth considering that in our first version of the study, we already used ~150 adult and ~60 postnatal animals. Nevertheless, taking into account that these experiments are laborious due to the complexity of both models of oligodendrocyte pathology, **in our revised study we addressed both the dose and the mode of administration** (see reviewer 2 point 2.3), performing a **new experiment with 30 adult animals**, and **also using** the pro-remyelinating drug **clemastine** to compare with our compounds. Considering that the doses used before were in the lower range of those previously reported (Sahdeo et al; 2014; Okazaki et al; 2018; Frye et al., 2018; Rossignol et al., 2021), **we tried oral administration by gavage** (theoretically better controlling the dose administered) at two concentrations, the same concentration (to compare with the drinking water treatment) **and a higher dose**. This corresponded to 2X-dose for dyclonine/Sm5 (10mg/kg to be at the highest range used in adult mice) and 10X-dose of leucovorin/Sm11 (5mg/kg, to be in the higher range used in humans).

After two days of gavage, we found clear signs of stress in all groups, including weight loss. Therefore, from the third day onward, we came back to the posology through the drinking water, with mice showing no more signs of stress and weight loss. Immunofluorescent analysis at 7 days post-lesion, to compare with our previous experiment, showed similar pro-oligodendrogenic effects at both doses tested (Fig. 7 extended data 1d-h), and confirmed our previous experiment (Fig. 7f-i). Moreover, while clemastine induced an increase in iOL2 density similar to dyclonine and leucovorin (Fig. 7 extended data 1h), **both leucovorin and dyclonine increased the density of proliferating OPCs** (Fig. 7m, o), **an effect that was absent for clemastine** (Fig. 7o). We have added this data in **lines 352-361**.

In the context of the hypoxia model, the intranasal administration of these compounds immediately following hypoxia is a strategy representing a noninvasive technique for drug administration, that was already demonstrated to impact brain cell behavior and eventually their proliferation (Scafidi et., 2013; Azim et al., 2017). With this treatment, we showed robust increase in the number of proliferating OPCs, progenitors fated to oligodendroglia, and an increase in the number of myelinating OLs in the leucovorin/Sm11 treatment (Fig. 6). Of course, this is without any intention to use intranasal administration in human preterm babies, which receive intravenous administration for most treatments. Our follow up project, NeoReGen, utilizes different mouse models of preterm brain injury to assess in more detail the translational capacity of leucovorin and dyclonine to foster brain repair in the context of early preterm brain injury.

3. The link to microglia, particularly the change in phagocytic / pro-regenerative microglial numbers, is intriguing but also needs to be strengthened. Did the authors look at the presence of myelin debris (e.g. dMBP staining)? Were changes in inflammatory signals observed?

In agreement to the reviewer points, to complement our study of microglial inflammatory, phagocytic and regenerative profiles, we now provide the quantification for myelin debris using dMBP staining (Shen et al., 2021) at 7 days post-lesion (dpl) showing that both leucovorin/Sm11 and dyclonine/Sm5

treatments reduce the amount of myelin debris in the lesion area and increase the proportion of dMBP phagocytosed by CD68⁺ microglia (Fig. 9a-e). With respect to changes in inflammatory microglial profiles, using iNOS and Cox2 as pro-inflammatory markers, we now show that both leucovorin- and dyclonine-treatments strongly reduce the inflammatory microglial profile at 7 dpl (Fig. 9f-h), in parallel to the increase in Arg1⁺ pro-regenerative microglia profile (Fig. 9i, j). Altogether these results strongly support that leucovorin and dyclonine promote/accelerate the lesion repair also by inducing microglia beneficial activities, including a faster clearance of myelin debris (dMBP by CD68⁺ cells) and an earlier transition microglial profiles from inflammatory (iNOS⁺/Cox2⁺) towards pro-regenerative (Arg1⁺). We have now added this data in Figure 9 and lines 394-403.

In Fig 7O, it also appears that lesion size might vary. Was this examined?

According to the reviewer comment, we reconstructed the lesion volume at 10 dpl to assess the impact of compounds' treatment on the lesion size finding no significant reduction of the lesion volume upon clemastine-, dyclonine-, and leucovorin-treatment, likely due to lesion variability. We now provide this data in Fig. 8c and in the main in line 371.

4. The bioinformatics data analysis needs clarification:

4a. The authors mention that raw sequencing data was downloaded from GEO, but it appears that the authors applied a microarray data processing pipeline. Can this be clarified?

According to the reviewer comment, we now clarify this in the methods in lines 559-565. We also now provide the bioinformatic analyses of the expression microarrays deposited in GEO of our previous publications (Azim et al., 2015; Azim et al., 2017) and detailed below.

4b. In addition, no details were documented about how the authors applied Seurat for single-cell RNAseq data analysis and for bulk RNAseq analysis. For instance, how did the authors correct for different batches for data integration? How many cells were integrated? What statistical method was used for differential analysis? Did the authors use adjusted p-value or a raw p-value as cutoff?

To obtain an oligodendroglial transcriptomic signature, we used expression data already processed by the respective authors from different sorts of datasets (i.e., expression microarrays, bulk-RNA-seq, and single cell RNA-seq datasets; see methods corresponding to all cited papers). For most datasets, the supplementary tables provided were used to select genes with enriched expression in each cell type (p-value < 0.05) or differentially expressed genes between different cell-subtypes/stages/clusters (>= 1.5-fold change compared to other cell types). As mentioned above, Methods table 2 summarizes all this information. We now also provide the script for the microarray analyses (dNSCs_dTAPs_Azim et al.R and OLglia_Zhang et al.R) and the provide a more detailed description in the Methods section *Data processing and oligodendroglial gene enrichment* in lines 555-601.

Minor Points

1. The sentence in the abstract "An unmet challenge is the discovery of medications presenting convincing pro-oligodendrogenic activity for the eventual clinical treatment of these pathologies" is somewhat misleading as there are several compounds currently being tested (e.g. clemastine, analogs of thyroid hormone) that promote OPC differentiation and remyelination. Of note, this is nicely discussed in the introduction.

According to the reviewer suggestion we have modified this sentence of the abstract to: [‘No medication presenting convincing repair capacity in humans has been approved for these pathologies’]

2. Figure 3 – Why was 750nM chosen as the dose to test all compounds (also 250 and 500nM in Figure 3, extended data 1)?

As mentioned in our answer to reviewer#1 question 6, we used NPC cultures to perform a systematic *in vitro* testing the pro-oligodendrogenic activity **dose effect** of our **11 selected compounds** at **3 different concentrations** (250 nM, 500 nM, and 750 nM) based on the previous reports (Mei et al., 2014; Najm et al., 2015; Eleuteri et al., 2017) and in **4 replicated experiments** (Figure 3 Extended data 1). Whole coverslip automatic quantifications of Sox10+ cells indicated that all selected compounds presented the strongest effects to promote Sox10+ oligodendroglia at 750nM. Given the results obtained in this extensive *in vitro* experiment, we carry out following *in vitro* and *ex vivo* experiments at this optimal dose (750 nM). We have made this point clearer in **lines 191-197**.

3. Figure 5 – CaBP staining is in green, but the CaBP+ axons are shown in white?

The overlap between CaBP+ axons labeled **in green** and MBP+ myelin labeled **in magenta**, gives the overlapping white color of ‘myelinated axons’ quantified in the myelination index. We now provided higher magnification pictures illustrating this point in Fig. 5 extended data 1b.

4. For the slice cultures, it would be helpful to have another metric of myelination, perhaps Caspr staining to axons, which would mark compact myelin and not be confounded by MBP+ oligodendrocytes on axons.

In many papers (including Yuen TJ, Silbereis JC et al., Cell 2014 cited by the reviewer and the method book chapter entitled *Investigating demyelination, efficient remyelination and remyelination failure in organotypic cerebellar slice cultures*, Gorter et al., 2022; DOI: 10.1016/bs.mcb.2021.12.011) myelination in explant cultures is quantified by the overlap between the axonal and the myelin segments. We now also provide higher magnification pictures making this more visible in Figure 5 extended data 1b, c). It would be very challenging to optimize a whole explant automatic quantification of Caspr+ paranodes (as we do in our experiments for all the markers used), and will not reinforce the messages of the paper. Therefore, given that it is mentioned by the reviewer as a minor point, we have concentrated our efforts and limited the number of animals sacrificed to address the major points. Note that we have now provided immunofluorescence and electron microscopy data showing that at 10 days post-lesion, both leucovorin- and dyclonine-treatment increased the number of newly-formed myelinating oligodendrocytes in the lesion area and, that leucovorin-treatment significantly increased the myelin thickness of axons larger than 1 μ m (Figure 8).

It is intriguing that clemastine promotes OL differentiation, but not myelination, while T3 does not promote differentiation, but promotes myelination.

It is noteworthy that both T3 and clemastine treatments increase the differentiation and myelination indexes (Figure 5c, e), even though the experimental variability makes that these differences reach significance in one or the other case. Indeed, as shown in the graphic below, g-test between T3 and clemastine indicates that the differences between their sample values are non-significantly different

(n.s.) in either differentiation index (Mann-Whitney test; p-value = 0.1893) or myelination index (Mann-Whitney test; p-value = 0.0907).

5. Discussion on why Sm2 was not selected to move forward to in vivo studies would be helpful. Is it simply that it was ranked #3 (as shown in Figure 5, extended data 1), or was there additional rationale?

Sm2/Heptaminol is myocardial stimulant and vasodilator, thus having a risk of side effects in the context of preterm brain injury, what made us not select it in the top two compounds. Its applications in the context of adult MS lesion repair will require future *in vivo* animal studies.

6. The manuscript could benefit from proofreading throughout (some grammar errors and typos). We thank the reviewer for his/her comments, and we have carefully paid attention to this aspect in our revised manuscript checked by English native-speakers.

Reviewer #3 (Remarks to the Author):

This manuscript tackles an important aspect of debilitating diseases such as newborn brain injury and multiple sclerosis, that of identifying therapeutic agents which promote remyelination in injury. Several other studies have used cell screening approaches in vitro and in vivo to identify compounds which promote remyelination. **The novelty of this paper lies in the demonstration of an informatics approach to predict compounds** which will induce oligodendrogenesis. The authors use this approach to identify predicted compounds which will promote remyelination after injury, and use two of them in vivo to promote injury repair. **I like the idea of this novel informatics approach a lot,** but have quite a few issues with its execution:

We thank the reviewer for the enthusiasm in the clinical relevance of our study and the novelty of our pharmacogenomic approach. We answer below to the questions/points raised.

1) It seems unclear from the description **how SPIED is predicting bioactive molecules** which induce transcriptional changes that induce oligodendrogenesis. A more thorough description of how this tool is used would be useful.

Following the suggestion of the reviewer, we have added a paragraph to the Methods section entitled *Pharmacogenomics using SPIED platform* in lines 603-613.

2) The main novelty of this paper is the informatics approach used. The authors discuss different gene sets for different processes of oligodendrocyte biology, specification, proliferation, migration, survival, differentiation, myelination. But it seems as if they lump all processes together to identify drugs that are 'oligodendrogenic'?

As indicated in the introduction, we chose 'a more comprehensive strategy' to find **drugs acting on several processes of oligodendrogenesis**, thus ranking and selecting drugs by the total score (sum of individual process' scores) which explain our results showing that dyclonine/Sm5 and leucovorin/Sm11 can act in different cell types and promote different oligodendroglial processes: (1) NPC differentiation into oligodendroglia, (2) OPC proliferation, (3) OPC differentiation, and (4) accelerates microglial phagocytosis and its faster transition from pro-inflammatory to pro-regenerative microglial profiles. We have state this aspect more clearly as follows:

Introduction lines 91-97: ['While most previous studies have followed a gene/pathway candidate approach, here **we aimed at using a more comprehensive strategy** (Fig. 1 extended data 1). To this end, we developed an *in-silico* approach combining both unbiased identification (i.e., transcriptomes) and knowledge-driven curation of transcriptional changes associated with oligodendrogenesis (i.e., OligoScore). We then identified small bioactive molecules (compounds) **capable of globally mimicking this transcriptional signature.**'],

We have now made this point clearer in the abstract and first paragraph of the discussion, as follows:

Abstract: ['Here, we present a pharmacogenomic approach leading to the identification of small bioactive molecules with a **large pro-oligodendrogenic activity**, selected through an expert curation scoring strategy (OligoScore) **of their large impact on transcriptional programs controlling oligodendrogenesis and (re)myelination**']

Discussion lines 413-420: ['While most previous studies have followed a gene/pathway candidate approach, **here we used a more global strategy to fill this gap**. Leveraging transcriptomic datasets through a pharmacogenomics analysis and developing an expert curation of genes previously involved in oligodendroglial biology (provided as a resource for the scientific community: [OligoScore, https://oligoscore.icm-institute.org](https://oligoscore.icm-institute.org)), we identified and ranked novel small bioactive molecules (compounds) **fostering transcriptional programs associated with various aspects of oligodendrogenesis**, including OPC proliferation, differentiation, and (re)myelination.']

Were compounds assessed that effect specific individual processes?

Yes. Figure 2B illustrates the genes in each process of oligodendrogenesis used to query SPIED, obtaining a total of 393 different compounds, with 156 of them overlapping with those obtained using the oligodendroglial transcription signature (Figure 2c). For these 156 selected compounds, we now provide the processes identifying each compound (**Supplemental table 7, column C**). It is worth noting that, given that each compound is scored by their curated genes involved in each process of oligodendrogenesis (columns G to M), most of the compounds on the upper part of the list (with higher total scores) were identified using the curated genes of several (2 or 3) oligodendrogenic processes (as indicated in column C). As mentioned above, in our study we focused on compounds having a large pro-oligodendrogenic activity (top of the list), and indeed, our experimental data validates this strategy given that using different experimental models, we demonstrate the capacity of Sm5/Dyclonine and Sm11/Leucovorin to increase oligodendroglial specification (Fig. 3c, g and Fig. 6f), OPC proliferation (Fig 6d, 7m,n), OPC differentiation (Fig. 4d, 5c, 7h,i), and the number of myelinating OLs (Fig. 5e and Fig. 8).

3) A somewhat similar point, of the compounds selected, which transcriptional hubs are activated by each and which individual processes are predicted to be affected by this?

Even though these questions beg for follow up studies, according to the reviewer request, we now provide the list of 252 up- and down-regulated transcriptional hubs associated with the genes regulated by dyclonine/Sm5 (**Supplemental table 9**) and leucovorin/Sm11 (**Supplemental table 10**), together with the enriched biological process (Gene Ontology) associated with them. It is worth noting that the biological processes associated with correlative gene hubs are mainly related to oligodendroglial development, while those associated with anti-correlative genes hubs are mainly related to immune and inflammation processes. We also provide now the OligoScore analysis of correlative hub genes regulated by dyclonine/Sm5 (**Supplemental table 11**) and leucovorin/Sm11 (**Supplemental table 12**) indicating the processes of oligodendrogenesis affected by the curated hub genes (i.e., those with already known function in oligodendrogenesis).

4) In both in vivo hypoxia and lysolecithin models, the chosen compounds are described as increasing OPC proliferation as well as the number of differentiated OL. It is difficult to understand how a drug is affecting both of these two very different processes. It is important to describe gene hubs affected by the drugs and how this leads to effects on proliferation and differentiation.

According to the reviewer's suggestion, we investigated how the hub genes regulated by Sm5 (Supplemental table 11) and Sm11 (Supplemental table 12) influence the processes of proliferation and differentiation by querying OligoScore for the known activities of these genes. We found 6 genes promoting proliferation, with 5 of them commonly regulated by Sm5 and Sm11 (*Cntnn1*, *Olig2*, *Ascl1*, *Myt1*, and *Sox2*) and many genes promoting differentiation (18 for Sm11, and 14 for Sm5; see Supplemental tables 11 and 12). Interestingly, four of these genes encode for key transcription factors (*Olig2*, *Ascl1*, *Sox2*, and *Myt1*) promoting both proliferation and differentiation programs (see references associated to each gene and process in OligoScore, thus allowing a balance between both cell fates (proliferation and differentiation) in OPCs. These results explain, at least in part, how Sm5 and Sm11 promote parallel OPC proliferation and differentiation. We discuss this point in the discussion section lines 431-436.

It is also worth mentioning that the current evidence based in long term *in vivo* live-imaging of adult OPCs in physiological or demyelinating conditions, suggests that OPCs start differentiation without previous division, with the 'hole' left by their differentiation being 'filled' by the proliferation of neighboring OPCs (Hughes et al., 2013, DOI: 0.1038/nn.3390). This is also supported by single cell transcriptomics showing different subsets (/clusters) of OPCs (see Marques et al., 2016, 2018; Spitzer et al., 2018, DOI: 10.1016/j.neuron.2018.12.020). Therefore, it is conceivable that compounds fostering the metabolism of OPCs, like leucovorin promoting the one-carbon metabolism, can on the one hand promote the differentiation those OPCs in a state prompt to differentiate, while on the other hand, induce OPCs in different state to proliferate more actively.

5) The most confusing part of the manuscript comes at the very end. The point of the paper is to identify compounds which affect oligodendrocyte transcriptional networks, but then at the very end the authors seem to be suggesting that the compounds they chose also alter a microglial signature in vivo in injury, raising the possibility that the promotion in remyelination has nothing to do with the effects on OPCs but due to altered microglia.

As stated above, our strategy aimed to select compounds having a large pro-oligodendrogenic activity, based on their regulatory impact on several transcriptional programs. It is thus not surprising

that the selected compounds, particularly sm5/dyclonine and sm11/leucovorin studied in more detail *in vivo*, can also impact other cell-types than oligodendroglia, including neural progenitor cells (NPCs) and microglia. During our study we demonstrated: (1) using neonatal neural progenitor cells' cultures (thus in the absence of microglia), the capacity of all selected compounds (Sm1 to Sm11) to foster differentiation of (NPCs) into oligodendroglia (figure 3); (2) using purified OPC cultures (absence of microglia), the direct capacity of Sm5 and Sm11 to promote OPC differentiation (figure 4); and (3) the *in vivo* capacity of Sm5 & Sm11 to foster neonatal NPC differentiation into oligodendroglia (figure 6, a process never described before to involve microglia). Therefore, these results demonstrate a direct activity of Sm5 and Sm11 in NPCs, OPCs, and likely differentiating OLs to promote oligodendrogenesis. Thus, the fact that Sm5 and Sm11 have effects in other cell types, such as microglia, is also not unexpected given that we selected the also for their: (a) previous reports of their anti-inflammatory activities (Rufini et al., 2022; Cianciulli et al., 2016; Tommy et al., 2021), (b) some of their pharmacogenomics' target genes (e.g., Sm11/leucovorin downregulates *IFNG*, the gene coding for the inflammatory cytokine Interferon gamma), and (c) their abovementioned anti-correlative genes hubs are mainly related to immune and inflammation processes. In this line of argument, the anti-inflammatory effect of Sm11 is discussed in **lines 510-517**.

Point by point response to reviewers' comments (2nd round)

Response in blue.

We would like to express our gratitude for the valuable feedback provided by the reviewers. With their insightful comments and our broader analyses, our results more strongly support and further extend the pre-clinical aims of the study, i.e., demonstrate the pro-oligodendrogenic and anti-inflammatory activities of leucovorin and dyclonine, two FDA-approved compounds, leading to brain repair in two models of oligodendrocyte pathologies, thus significantly widening its potential for future clinical impact.

Reviewers' comments:

Reviewer #2 (Remarks to the Author):

Thank you to the authors for their revision and additional experiments. While the authors conducted numerous additional experiments and useful discussion, which help strengthen their claims, there are still points that are central to their claims that need to be strengthened.

1) The authors main aim is to identify compounds that promote repair that can translate to pathologies such as early preterm-birth brain injury (PBI) and adult multiple sclerosis (MS). They have done extensive work to show the enhancement of proliferation and oligodendroglial fate, as well as differentiation. However, as the ultimate aim is to promote myelination or remyelination, this still needs to be strengthened.

a. The fate mapping experiments (Figure 8) add to the story, but still do not address the myelination component. For instance, there could be many mature oligodendrocytes, which does not translate into myelin formation. The claim that these are (re)myelinating oligodendrocytes is not supported with data.

We would like to express our disagreement with the reviewer's statement. Following previous suggestions from the reviewer, to strengthen this aspect of the study, we performed two relevant and demanding experiments: (1) a new experiment using 30 additional adult mice to genetically label and quantify the newly formed OLs in the lesion area, using the pro-myelinating compound clemastine as positive control; (2) an experiment using 20 adult mice to perform EM analysis (see next point).

In our immunofluorescent analysis, we used GSTp, that together with ASPA (Pan et al., 2020; <https://doi.org/10.1038/s41593-019-0582-1>), is a marker labeling the cytoplasm of mature/myelinating oligodendrocytes, thus allowing quantification of OL numbers, as previously demonstrated by several studies (Zhou et al., 2021; <https://doi.org/10.7554/eLife.60467>; Hassel et al., 2023; <https://doi.org/10.1002/glia.24373>; Tansey & Cammer, 1991; <https://doi.org/10.1111/j.1471-4159.1991.tb02104>). Moreover, we provided direct evidence that GSTp⁺ cells are indeed myelinating OLs. First, in the context of the postnatal brain, we show that GSTp⁺ cells have the typical morphology of myelinating OLs visualized by the GFP reporter-labeling in myelin segments/internodes (Fig. 6 extended data 1e). Second, in the context of the adult de/remyelination model, where we genetically labeled OLs generated from adult OPCs (by administering tamoxifen before lesion induction to *Pdgfra-CreERT/Rosa26^{flox-stop-flox-YFP}* mice), we used GSTp⁺ to immunodetect myelinating OLs, showing that many GSTp⁺ cells presented the typical alignment of myelinating OLs along axons in the corpus callosum (Fig. 8d). Moreover, at the same time, we used *Bcas1* to immunodetect pre-myelinating/immature OLs (Fard et al., 2017), finding no overlap between *Bcas1*⁺/YFP⁺ cells, presenting a large cytoplasm and long processes typical of pre-myelinating/immature OLs (see Fig 8d, high mag panels), and GSTp⁺/YFP⁺ cells, having a small rounded cytoplasm typical of myelinating OLs (Fig. 8d), strongly supporting that GSTp⁺/YFP⁺ cells are newly-formed (YFP⁺ cells) myelinating OLs.

Finally, we would like to highlight that to the best of our knowledge, we did not find papers providing evidence of the existence of mature non-myelinating OLs in the LPC or cuprizone de/remyelinating murine models. Indeed, the heterogeneity of mature OLs shown by single-cell transcriptomic analyses in adult mice (Marques et al., 2016) and in the brain of human MS patients and healthy controls (Jakel et al., 2019) has not been shown by follow-up studies to correspond to non-myelinating oligodendrocytes. For example, time-lapse analysis in the cuprizone model, using transgenic mice reporting with fluorescence OL myelin segments, identified regenerated oligodendrocytes in the lesion area by the mature form reached 12–14 days after the first appearance, all of them being myelinating oligodendrocytes (Orthmann-Murphy et al., 2020, DOI:

10.7554/eLife.56621). Also, in a recent study performing combination between spatial transcriptomics and electronic microscopy analysis in the LPC de/remyelinating model (Androvic et al., 2023, DOI: 10.1038/s41467-023-39447-9), the authors identified four mature OL subtypes/clusters (Oligo 1-4), with Oligo 3 and 4 clusters representing injury-responding states of mature oligodendrocytes, with Oligo 3 OLs expressing marker genes of previously defined disease-associated OLs in aging, AD and MS models, and Oligo 4 OLs expressing a battery of interferon-stimulated genes, and thus called interferon responsive OLs. It is worth noting that the study did not provide any evidence or suggestion that one of these mature OL subtypes corresponded to mature non-myelinating OLs.

To make this point clearer in lines 367-373, we have now expressed it as follows: [‘and at 10 dpl identified by immunofluorescence newly formed OLs (YFP⁺ cells) being either Bcas1, a marker of immature/pre-myelinating OLs (Fard et al., 2017), or GST π , a marker restricted to myelinating OLs (Zhou et al., 2021). Interestingly, we did not find overlap between YFP⁺/Bcas1⁺ cells and YFP⁺/GST π ⁺ cells, suggesting that Bcas1 and GST π indeed identify immature/pre-myelinating and mature/myelinating OLs, respectively.’]

Tansey FA & Cammer W. A pi form of glutathione-S-transferase is a myelin- and oligodendrocyte-associated enzyme in mouse brain. JNeurochem 1991. DOI: 10.1111/j.1471-4159.1991.tb02104

Pan, ..., Jonah R. Chan & Mazen A. Kheirbek. Preservation of a remote fear memory requires new myelin formation. Nat Neurosci. 10 Feb 2020. DOI: 10.1038/s41593-019-0582-1

b. Did the authors examine a quantitative or semi-quantitative measure of remyelination in their EM study? G-ratios likely are not a very useful metric in the corpus callosum.

As mentioned above, we used 20 additional adult animals to perform the EM analysis, showing that myelinating OLs in the lesion area, identified by their nucleus of round- or oval-shape having densely packed chromatin and processes wrapping around myelinated axons, were more abundant (**semi-quantitative measure**) in animals treated with leucovorin, dyclonine, and clemastine compared to vehicle-treated animals, as illustrated in Fig. 8e, paralleling the results quantified using immunodetection of GSTp⁺/YFP⁺ OLs (Fig. 8b, d). We now added a supplemental figure (**Figure 8 extended data 1**) with higher magnification micrographs illustrating newly formed (re)myelinating OLs (mOLs) in the lesion area, identified by their typical ultrastructural traits (i.e., oval-shape nucleus having densely packed chromatin with several heterochromatin spots and large cytoplasm in continuity with axons presenting compact myelin ultrastructure, red arrows) in animals treated with our compounds.

We also used EM to quantify myelinated axons in the lesion area (identified by cellular and extracellular disrupted structures; see Fig. 8e) finding that almost all axons were (re)myelinated, as illustrated in panels of Fig. 8f. Despite our early timing of analysis for the remyelination of the lesion (10 days post-lesion), we could detect a tendency of increased myelin thickness (decrease in g-ratios; Fig. 8g; **semi-quantitative measure**) in axons from all compound-treated animals compared to vehicle-treated. Moreover, we found a statistical increase in myelin thickness (lower g-ratio) of axons with a diameter larger than 1mm in leucovorin-treated animals, compared to the vehicle (Fig. 8h; **quantitative measure**). These results could be interpreted as compounds accelerating the process of lesion remyelination. Nevertheless, we only state: [‘suggesting an increased remyelination (more wrapping) induced by the compound treatment’ [...] ‘Altogether, these results show that, similar to clemastine, leucovorin and dyclonine promote OL differentiation and remyelination *in vivo* in the context of adult brain demyelination, with leucovorin showing the strongest effect’.]

To be even more careful in our conclusions, we propose to rephrase this in lines 387-388 as follows: [‘promote OL differentiation and **thus remyelination. The possibility of leucovorin and dyclonine directly inducing myelin formation would require further investigation.**’]

c. Figure 6f, h, j – was the difference between Hx vs. Hx + Dycl. statistically significant? It is not shown, but the text is inconsistent in the results (mentioning a rescue and then more mitigated effects).

Both Fig. 6h and 6j show that the statistical reduction in CC1⁺ cells and GSTp⁺ cells between Hx vs. Nx is ‘rescued’ (non-significant difference with normoxic group, n.s.) in hypoxic animals treated with dyclonine and leucovorin. Only in the case of leucovorin, the data reach a statistical difference between Hx vs. Hx + Leuc, both in Fig. 6j, 6h, and 6l, as also shown in the heatmap representations depicting the automatic quantifications of the density of GSTp⁺ cells (myelinating OLs) in different brain regions (Fig. 6k).

According to the reviewer's comment we propose to change the text in main lines 293-295 as follows:

[‘their differentiation into CC1⁺/Olig2⁺ OLs was impaired by hypoxia but **this difference with the normoxic group was rescued by both dyclonine and leucovorin treatments, with leucovorin also reaching statistical difference with the hypoxic group**’]. In lines 298-299 [‘Quantification of Olig2⁺/GSTπ⁺ cell density revealed a marked effect of hypoxia on OL maturation that was fully rescued by leucovorin treatment, ~~while dyclonine showed more mitigated effects.~~’]

We have also changed the figure legend of Fig. 6 accordingly: [‘(h) showing that dyclonine and leucovorin rescue the **reduced density of OL differentiation differentiating OLs** (CC1⁺/Olig2⁺ cells) ~~following induced by~~ neonatal chronic hypoxia within the cortex at P19 to **the density found in the normoxic group, with leucovorin reaching statistical difference with the hypoxic group**’.]

It is worth noting that in the section named ‘limitations of the study’, we already discuss this as follows: [‘the increase in OPC proliferation and numbers mediated by dyclonine in the hypoxia model do not rescue the density of myelinating OLs, contrasting with its effects in adult brain de/remyelination, calling for a better understanding of the environmental differences of each pathological model, such as microglial involvement.’]

d. The authors conclude that the compounds have the capacity to cross the BBB, but no data are shown to directly measure compound concentration in the brain to support this conclusion?

It should be considered that we selected leucovorin and dyclonine given that they were FDA-approved compounds with previous publications demonstrating the capacity to cross the BBB. Indeed, for leucovorin/Sm11, the crossing of the BBB is supported by several publications both in animal models and in humans, as mentioned in the discussion section lines 499-505: [‘Both leucovorin and folic acid can be transformed by different enzymes into 5-methyl-tetrahydro-folate, which efficiently crosses the BBB using the folate receptor alpha (FR α) transporter, and is thought to be the main active folate in the CNS (Scaglione and Panzavolta, 2014). Leucovorin has the advantage over folic acid of using other transporters present in the choroid plexus to get into the CNS, i.e., the proton-coupled folate transporter (PCFT) and the reduced folate carrier (RFC) (Mafi et al., 2020): (...) leucovorin has been used to treat epileptic patients having mutations in the folate receptor alpha gene, FOLR1 (Mafi et al., 2020)’]. We also give references supporting it in Table S8 column F, and in the file called *references BBB* (S8-6, S8-1, S8-2). For dyclonine/Sm5, we mention in the discussion lines 484-487 that [‘In a study of drug repositioning in the context of Friedreich Ataxia, dyclonine was found to confer protection against diamide-induced oxidative stress through binding to the transcription factor NRF2’].

To make this point clearer, we have now added the original reference of this study (Sahdeo et al., 2014; doi: 10.1093/hmg/ddu408), already cited in the methods section, in which mice and human patients are treated with oral administration of dyclonine showing improvement from symptoms associated with Friederich Ataxia. We also provide the original reference of crossing the BBB in S8-21 in the file *references BBB*.

Finally, it is beyond the scope of this large screening study to generate data providing compound concentration in the mouse brain, which should be done in follow-up studies.

2) Thank you for adding Methods Table 2 to summarize the expression datasets included in this study and the statistics for selecting DEGs. However, the single-cell data processing documented in lines 597-600 is still confusing. How did the authors apply both ‘min genes=500 and max genes<200’ filtering conditions simultaneously? Why are the authors interested in cells with high mitochondrial percentage (>10%)?

a. The following link may help with general guidance: https://satijalab.org/seurat/articles/pbmc3k_tutorial.html

b. I assume the authors used the first 20 PC dimensions for UMAP or tSNE clustering with a clustering reduction of 0.9? For the 1,598 DEGs identified, which statistical method was used here?

(using the following criteria: min.cells = 10; min.genes = 500; number of UMI >5100; max genes <200; mitochondrial percentage >0.10, analysis in 20 dimensions with a resolution of 0.9)

We thank the reviewer for spotting this mistake. We have corrected this paragraph in lines 602-612 as follows: [‘using Marques and colleagues’ (2016) supplemental table ‘aaf6463 Table S1’, sheet ‘specific genes’ we pooled all genes except those of VLMC column obtaining 532 unique genes. To extract more information, raw data from 5072 cells were processed with the Seurat package v2 (Stuart et al., 2019) using the following criteria: min.cells = 10; min.genes = 500; mitochondrial percentage <0.10, 20 dimensions to find neighbors, a resolution of 0.9 to find clusters, selecting the top150 genes from each cluster from

FindAllMarkers function, identifying 1598 DEGs (Methods Table 2).']. We have modified accordingly the text in Methods Table 2 and provide the R script ('Marques 2016 gene selection.R').

Reviewer #3 (Remarks to the Author):

I appreciate the authors responses to my questions, and feel now they have addressed all the concerns.

We thank the reviewer for the useful comments that have allowed us to improve the quality of the study.